**Measuring rates of present-day relative sea-level rise in low-elevation coastal zones: A**
**critical evaluation**
Molly E. Keogh[1] and Torbjörn E. Törnqvist
Department of Earth and Environmental Sciences, Tulane University, 6823 St. Charles Avenue,
New Orleans, Louisiana 70118-5698, USA
[1]Corresponding author: mkeogh@tulane.edu






## 1. ABSTRACT

Although tide gauges are the primary source of data used to calculate multi-decadal to century-scale rates of relative sea-level change, we question the usefulness of tide-gauge data in rapidly subsiding low-elevation coastal zones (LECZs). Tide gauges measure relative sea-level rise (RSLR) with respect to the base of associated benchmarks. Focusing on coastal Louisiana, the largest LECZ in the United States, we find that these benchmarks ($n = 35$) are anchored an average of 21.5 m below the land surface. Because at least 60% of subsidence occurs in the top 5 m of the sediment column in this area, tide gauges in coastal Louisiana do not capture the primary contributor to RSLR. Similarly, Global Navigation Satellite System (GNSS) stations ($n = 10$) are anchored an average of >14.3 m below the land surface and therefore also do not capture shallow subsidence. As a result, tide gauges and GNSS stations in coastal Louisiana, and likely in LECZs worldwide, systematically underestimate rates of RSLR as experienced at the land surface. We present an alternative approach that explicitly measures RSLR in LECZs with respect to the land surface and eliminates the need for tide-gauge data in this context. Shallow subsidence is measured by rod surface-elevation table–marker horizons (RSET-MHs) and added to measurements of deep subsidence from GNSS data, plus sea-level rise from satellite altimetry. We show that for a LECZ the size of coastal Louisiana (25,000-30,000 km$^2$), about 40 RSET-MH instruments suffice to collect useful data. Rates of RSLR obtained from this approach are substantially higher than rates as inferred from tide-gauge data. We therefore conclude that LECZs may be at higher risk of flooding, and within a shorter time horizon, than previously assumed.

## 2. INTRODUCTION

In the current era of accelerated sea-level rise, accurate measurements of relative sea-level change are critical to predict the conditions that coastal areas will face in coming decades and beyond. Such measurements traditionally come from tide gauges, which provide the longest available instrumental records of relative sea-level rise (RSLR). Some of the oldest tide gauges have records spanning 150-200+ years [e.g. Key West, USA (Maul and Martin, 1993); Brest, France; Świnoujście, Poland; New York, USA; and San Francisco, USA (Woodworth et al., 2011); and Boston, USA (Talke et al., 2018)]. Tide-gauge data have played a central role in calculations of global sea-level rise (e.g. Gornitz et al., 1982) and they continue to do so today (e.g. Church and White, 2011; Church et al., 2013; Hay et al., 2015).

Tide-gauge data are also heavily relied upon to evaluate the vulnerability of low-elevation coastal zones (LECZs) (e.g. Syvitski et al., 2009; Nicholls and Cazenave, 2010; Kopp et al., 2014; Pfeffer and Allemand, 2016). LECZs include large deltas and coastal plains that often have accumulated thick packages (tens of meters or more) of highly compressible Holocene strata and are the home to some of the world's largest population centers (e.g. Tokyo, Shanghai, Bangkok, Manila) that are increasingly at risk due to RSLR. At the regional level, tide-gauge data have been used to study a variety of spatially variable processes. For example, in coastal Louisiana, the largest LECZ in the United States, tide-gauge data have been used to

measure land subsidence (Swanson and Thurlow, 1973), the acceleration of RSLR (Nummedal, 1983), multi-decadal rates of subsidence and RSLR (Penland and Ramsey, 1990), and the impact of fluid extraction on RSLR (Kolker et al., 2011).

The Permanent Service for Mean Sea Level (PSMSL; http://www.psmsl.org; Holgate et al., 2013) maintains records for nearly 2000 tide gauges globally, including five in coastal Louisiana: Eugene Island (data from 1939-1974), Grand Isle (1947-present), South Pass (1980-1999), Shell Beach (2008-present) and New Canal Station (2006-present). In many parts of the world, however, tide gauges with long, continuous records are few and far between. As a result, many studies of RSLR rely on tide-gauge records that are too short (longer than 50 years is preferable but at least 30 years is necessary to filter out natural variability due to phenomena such as storms, El Niño-Southern Oscillation cycles, changes in the orbital declination of the moon, shifts in ocean currents, and atmospheric pressure variability; Pugh, 1987; Douglas, 1991; Shennan and Woodworth, 1992), are from inappropriate locations (e.g. outside of the area being studied), or both. For example, of the 32 tide gauges used by Syvitski et al. (2009), 21 were located outside the delta of interest, 11 had records of <30 years, and 8 had both shortcomings. Furthermore, subsidence rates are highly spatially variable, often increasing or decreasing 2- to 4-fold within short distances (a few km or less) as a result of subsurface fluid withdrawal and differential compaction, among other factors (e.g. Teatini et al., 2005; Törnqvist et al., 2008; Minderhoud et al., 2017; Koster et al., 2018; also see the review by Higgins, 2016). As a result, tide gauges provide limited information on subsidence rates beyond the instrument's immediate surroundings. Even if a tide gauge has a sufficiently long record and is appropriately located, it is critical to determine what processes the tide gauge is measuring, and what it is not measuring. In LECZs, this is commonly not straightforward.

Tide gauges measure RSLR with respect to a nearby set of benchmarks. Leveling campaigns are conducted regularly [for example, at least once every six months for National Oceanic and Atmospheric Administration (NOAA) tide gauges; NOAA, 2013] to account for any changes in the elevation of the tide gauge with respect to these reference points. Tide gauges are typically leveled using a benchmark designated as the primary benchmark; secondary benchmarks are used to assess the stability of the primary benchmark (NOAA, 2013).

Figure 1 shows a schematic of tide gauges and associated benchmarks in three contrasting environments. Along rocky coastlines, benchmarks are typically anchored directly onto bedrock that is exposed at the surface (Fig. 1a). A tide gauge in such a setting therefore measures RSLR with respect to the land surface. In contrast, benchmarks in LECZs are typically anchored at depth. In thin LECZs, which are defined herein as those with unconsolidated sediment packages <20 m thick, benchmark foundations typically penetrate the surficial layer of unconsolidated (usually Holocene) sediment and are anchored in the underlying consolidated (usually Pleistocene) strata (Fig. 1b). In thick LECZs, defined as possessing unconsolidated sediment packages that are >20 m thick, benchmark foundations are generally not sufficiently deep to reach the consolidated strata and are anchored within the unconsolidated sediment (Fig. 1c).

Regardless of the environment, all tide gauges measure changes in water surface
elevation with respect to the foundation depth of their associated benchmarks. As a result, tide
gauges with benchmarks anchored at depth do not account for processes occurring in the shallow
subsurface, above the benchmark foundation (Cahoon, 2015). For the purposes of this study, we
define the subsidence that occurs above a benchmark's foundation as "shallow subsidence"
(*sensu* Cahoon et al., 1995). Subsidence below a benchmark's foundation is termed "deep
subsidence". In coastal Louisiana, at least 60% of subsidence occurs in the shallowest 5-10
meters (Jankowski et al., 2017). Tide gauges with benchmarks anchored at depth do not record
this key component of RSLR (Cahoon, 2015). This issue was also recognized by Jankowski et al.
(2017) and Nienhuis et al. (2017), but neither study elaborated on this problem. Here, we present
a detailed assessment of benchmark information associated with tide gauges, followed by a
discussion of its implications as well as methods to remedy this issue.
In order to better understand the contribution of vertical ground motion to RSLR, tide-
gauge data are often used in conjunction with Global Navigation Satellite System (GNSS) data
(e.g. Mazzotti et al., 2009; Wöppelmann et al., 2009; Wöppelmann and Marcos, 2016; see also
the Intergovernmental Oceanographic Commission manuals on sea-level measurement and
interpretation, available at http://www.psmsl.org/train_and_info/training/manuals/). In LECZs,
GNSS stations are typically mounted on existing buildings or attached to rods that are driven to
refusal (i.e. the depth at which friction prevents deeper penetration; see International GNSS
Service station information at http://www.igs.org/network and National Geodetic Survey station
information at https://www.ngs.noaa.gov/CORS/) and record the deep subsidence that occurs
beneath their foundations. Similar to tide gauges, GNSS stations are nearly always anchored at
depth and thus face many of the same concerns: they do not record shallow subsidence that
occurs in the strata above the depth of their foundations.
Accurate measurements of RSLR are vital to predict the sustainability of world deltas
and for communities in LECZs to adapt to their changing coastlines. In this study, we investigate
the nature of tide gauge benchmarks and GNSS station foundations in coastal Louisiana and
assess the implications for measurements of RSLR and subsidence in LECZs worldwide. Re-
analysis of time series from tide gauges and GNSS stations is not the purpose of our study.
Instead, we present an alternative approach to measuring RSLR in LECZs where shallow
subsidence is determined using the rod surface-elevation table–marker horizon method [RSET-
MH; see Webb et al. (2013) and Cahoon (2015) for detailed descriptions of this method] and
deep subsidence is determined using GNSS data. Using the Mississippi Delta (a thick LECZ) and
the Chenier Plain (a thin LECZ) in coastal Louisiana as the primary study areas, we determine
benchmark foundation depths and the type of strata in which the foundations are anchored. This
allows us to determine which subsidence processes are measured by tide gauges and GNSS
stations and to evaluate their usefulness as recorders of RSLR. We then place our findings in the
context of LECZs worldwide. Our results suggest that tide gauges (and existing analyses of tide-
gauge data) in these environments may underestimate rates of RSLR as observed at the land
surface, and as a result, many LECZs may be at higher risk of submergence than previously
recognized.
3.   DATA AND METHODS
Relative sea level and subsidence data are abundant in the Mississippi Delta and Chenier
Plain, making coastal Louisiana an excellent target to assess methods of measuring RSLR.
Records for at least 131 operational or previously operational tide gauges in this region are
maintained by NOAA (https://tidesandcurrents.noaa.gov), the U.S. Army Corps of Engineers
(USACE; http://www.rivergages.com and Veatch, 2017), and the U.S. Geological Survey
(USGS; http://nwis.waterdata.usgs.gov). Although 37 of these tide gauges have records spanning
more than 30 years, many of their records are incomplete and have large data gaps. Many other
tide gauges in coastal Louisiana have short records; nearly half have time series <10 years and a
quarter are <2 years long (see Table S1 for information on all 131 tide gauges).
By means of exhaustive record combing of NOAA, USACE, and USGS archives,
benchmark foundation depths were determined for tide gauges located in the Holocene landscape
of the Mississippi Delta and Chenier Plain. Foundation depths were then compared to the local
elevation of the Pleistocene surface (with respect to the North American Vertical Datum of 1988,
NAVD 88; Heinrich et al., 2015). Because the land surface elevations at the tide gauge locations
are close to sea level, the elevation of the Pleistocene surface is essentially equivalent to its depth
beneath the land surface. When a tide gauge is associated with multiple benchmarks, the
benchmark with the deepest known foundation was used for this analysis. For comparison, the
analysis was repeated using primary benchmarks only.
A similar approach was taken to determine foundation depths of GNSS stations. GNSS
station information was compiled from Dokka et al. (2006) and Karegar et al. (2015). Of the 45
GNSS stations used for analysis by one or both studies, 17 are located in the Holocene landscape
of coastal Louisiana. GNSS station foundation depths were compared to the local depth of the
Pleistocene surface, similar to what was done for the tide gauges.
4.  RESULTS
The 131 tide gauges in coastal Louisiana were examined for benchmark information
(Table S1, Fig. 2). Benchmark foundation depths are available for only 35 tide gauges (Table 1),
including 31 maintained by NOAA and 4 maintained by USACE (see Table S1 for information
on all 131 tide gauges). Each of these NOAA tide gauges is associated with 3 to 11 benchmarks
(mean = 6 benchmarks), 77% of which have known foundation depths. The total number of
associated benchmarks is unknown for the USACE tide gauges. Benchmarks with known
foundation depths are typically mounted on steel rods driven to refusal. Benchmarks with
unknown foundation depths are typically mounted on concrete structures of a variety of types
(e.g. building foundations, bridge abutments, and seawalls). These concrete structures are likely
to have foundations that extend into the subsurface, but specific construction details are
unknown. It is important to note that an unknown foundation depth should not be interpreted as a
foundation depth of zero. The remaining 96 tide gauges (73% of the total) have no available
benchmark foundation information.
For tide gauges with available benchmark information, benchmark foundation depths
range from 0.9 to 35.1 m, with a mean of $21.0 \pm 5.4$ m and a median of 20.7 m. Deepest known
benchmarks are anchored an average of $21.5 \pm 7.4$ m below the ground surface, with a median
depth of 23.2 m. Comparing this mean to the mean foundation depth of primary benchmarks
$(21.4 \pm 3.9$ m, $n = 23)$, we find that there is no meaningful difference. Note that for 8 of these 23
tide gauges (35%), the primary benchmark is also the benchmark with the deepest known
foundation. The mean foundation depth for the shallowest known benchmarks is $17.3 \pm 7.0$ m.
When a tide gauge is associated with multiple benchmarks, the benchmark with the
deepest known foundation was used for this analysis. Figure 3 shows the location of tide gauges
in coastal Louisiana (circles) and the foundation depth of their associated benchmarks relative to
the local depth to the Pleistocene surface. The depth to the Pleistocene surface from the land
surface at tide gauge locations ranges from 5 to 142 m, with a mean of $47 \pm 34$ m and a median
of 44 m (Fig. 4). Thus, benchmark foundations are anchored an average of 26 m above the
Pleistocene surface. Only 11 of the 35 tide gauges (31%) have benchmarks anchored in
Pleistocene strata; the remaining 24 tide gauges (69%) have benchmarks anchored in Holocene
strata.
Of the 17 GNSS stations in coastal Louisiana, 10 (59%) have known foundation depths
(Table 2, Fig. 3). Information for all 17 GNSS stations in coastal Louisiana is available in Table
S2. Foundation depths of the 10 GNSS stations range from 1 to 36.5 m, with a mean of $>14.3 \pm$
$11.9$ m and a median of 14.9 m (Table 2). Note that for two GNSS stations only minimum
foundation depths are available; these minimum values are used in the analysis in order to
produce conservative results. At GNSS station locations, the depth to the Pleistocene surface
ranges from 10 to 78 m, with a mean of $39 \pm 20$ m and a median of 35 m (Fig. 4). Thus, GNSS
station foundations are anchored an average of 25 m above the Pleistocene surface. Only one of
the 10 GNSS stations (10%) is anchored in Pleistocene strata, whereas the remaining 9 GNSS
stations (90%) are anchored in Holocene strata. Figure 3 shows the location of GNSS stations in
coastal Louisiana (squares) and their foundation depth relative to the local depth to the
Pleistocene surface.
5.  DISCUSSION
*5.1. Implications for the interpretation of tide gauge and GNSS records*
In coastal Louisiana, foundation information for tide gauge benchmarks and GNSS
stations is often not available, essentially precluding the interpretation of resulting time series in
terms of rates of RSLR. Although many of the tide gauges listed in Table 1 are not useful for
RSLR analyses due to their short records, all of the benchmarks used for the present analysis are
currently published and considered stable. Furthermore, some of the tide gauges that currently
have short time series could become important in the future as their records become longer (e.g.
Shell Beach). Because all tide gauge benchmarks with known foundation information are

anchored at depth rather than at ground level, and most (91%) are anchored well below the land surface (>10 m), their interpretation is far from straightforward. Tide gauges with benchmarks anchored at depth measure deep subsidence plus the component of RSLR associated with changes in real (geocentric) ocean level, but do not capture shallow subsidence, often a dominant element of total subsidence in this region. Similarly, all GNSS stations are anchored at depth (60% are anchored >10 m deep) and also do not record shallow subsidence. Thus, tide gauges and GNSS stations in coastal Louisiana systematically underestimate the rates of local RSLR and subsidence, respectively.

Many tide gauges in coastal Louisiana have benchmarks that are mounted on existing concrete structures. The primary benchmark for the Grand Isle tide gauge, for example, is mounted on a seawall. Similar to tide gauges that measure RSLR with respect to a benchmark mounted on a steel rod driven to depth, the Grand Isle tide gauge produces a time series of RSLR with respect to the foundation of the concrete structure into which its primary benchmark is mounted. Although we were unable to acquire construction details for the seawall at Grand Isle, it is highly unlikely that it is simply resting on the land surface. We expect that the seawall foundation extends at least several meters into the subsurface in order to provide stability and protection to the adjacent Grand Isle Coast Guard station. Five other tide gauges also have primary benchmarks anchored on concrete structures: Caminada Pass, East Bay, Freshwater Canal Locks, Lafitte, and Martello Castle. Although all of these primary benchmarks are likely anchored at some depth below the surface, it is conceivable that their foundations are shallower than that of the deepest benchmarks (e.g. 19.8 m at Grand Isle). This may reduce the underestimation of the rate of RSLR measured by these tide gauges.

On the other hand, the RSET-MH data presented by Jankowski et al. (2017) suggest that shallow subsidence occurs dominantly in the uppermost 5 m in coastal Louisiana. Using data from 274 monitoring stations, Jankowski et al. (2017) calculated a mean shallow subsidence rate of $6.8 \pm 7.9$ mm yr$^{-1}$. Limiting this analysis to stations where the instrument is anchored in Pleistocene strata and the overlying (Holocene) strata are <5 m thick, we find a mean shallow subsidence rate of $6.4 \pm 5.4$ mm yr$^{-1}$ ($n = 55$). The similarity between these two numbers suggests that shallow subsidence is concentrated in the uppermost 5 m in this region. The implication would be that tide gauges with benchmarks anchored as little as 5 m below the surface would still not capture shallow subsidence and thus underestimate the rate of RSLR.

If a tide gauge benchmark is anchored in Pleistocene deposits, deep subsidence consists solely of subsidence within the Pleistocene and underlying strata (Fig. 1b). This scenario is common in LECZs with a relatively thin Holocene sediment package, such as the Chenier Plain. In the Chenier Plain, the Pleistocene surface subsides at a rate of ~1 mm yr$^{-1}$, yet the wetland surface is subsiding notably faster, at a rate of 7.5 mm yr$^{-1}$ on average (Jankowski et al., 2017). The remaining 6.5 mm yr$^{-1}$ of shallow subsidence occurs above the depth of local benchmark foundations and is typically not captured by tide gauges in this region.

In the case of a benchmark that is anchored in Holocene strata, deep subsidence also includes subsidence of the part of the Holocene sediment column that underlies the benchmark

foundation. This scenario (Fig. 1c) is common in LECZs with thick sediment packages such as the Mississippi Delta, and further complicates the interpretation of tide-gauge data. Compaction of deeper Holocene strata may result in an increase in the measured rate of RSLR when compared to tide gauges with benchmarks anchored in Pleistocene strata. However, tide gauges with benchmarks anchored in Holocene strata still record rates of RSLR that are considerably lower than what is seen at the land surface in the Mississippi Delta ($13 \pm 9$ mm yr$^{-1}$; Jankowski et al., 2017). For example, Kolker et al. (2011) and Karegar et al. (2015) calculated modern RSLR rates from tide-gauge data in the Mississippi Delta of ~3 mm yr$^{-1}$ (after adding the long-term rate of RSLR measured at Pensacola, Florida) and at least ~7 mm yr$^{-1}$, respectively.

Around the world, many LECZs have sediment packages that exceed 20 m in thickness, and some are as thick as 100 m or more (Table 3). Benchmarks in these areas are likely constructed in a broadly similar fashion to those in coastal Louisiana: either attached to rods driven to refusal or mounted on existing structures with non-negligible foundation depths. Tide-gauge benchmarks in The Netherlands, for example, are anchored 5-25 m deep (R. Hoogland, personal communication, 2018) and generally reach the Pleistocene basement except in areas where the Holocene sediment thickness is greatest (Table 4). Thus, conditions in The Netherlands are roughly comparable to those in the Chenier Plain of coastal Louisiana (and likely other "thin" LECZs): tide gauges do not capture the shallow subsidence component of RSLR, but because benchmarks are generally anchored in a relatively stable substrate they are easier to interpret than many of the tide gauges in the Mississippi Delta (and likely other "thick" LECZs) where benchmarks are essentially "floating" in the Holocene succession.

In LECZs globally, tide gauges likely underestimate the local rate of RSLR. A lack of reliable RSLR data will be increasingly problematic in several large deltas that are home to major population centers (e.g. Ganges-Brahmaputra, Song Hong, Yangtze, Mekong, Nile) and are experiencing rapid subsidence (Alam, 1996; Mathers and Zalasiewicz, 1999; Shi et al., 2008; Erban et al., 2014; Gebremichael et al., 2018). In these areas and in LECZs globally, people and infrastructure may therefore be even more vulnerable to flooding than previously recognized (e.g. Syvitski et al., 2009; Tessler et al., 2015).

Two studies that considered delta vulnerability on a global scale (Ericson et al., 2006; Tessler et al., 2015) are noteworthy because they did not depend on tide-gauge data. These studies determined RSLR by adding the historic rate of real (geocentric) sea-level rise to natural and anthropogenic subsidence data (Ericson et al., 2006) or by combining sea-level rise from satellite altimetry with subsidence estimates associated with fluid extraction (Tessler et al., 2015). While these approaches bypass the problems with tide gauges discussed above, they are also inherently limited by the need to characterize individual deltas by single metrics, by relying on measurements of global rather than local sea-level rise, and/or by not considering all major subsidence processes (notably shallow compaction). In the next section, we build on the recent study by Jankowski et al. (2017) to offer an alternative approach to measure RSLR in LECZs.

*5.2. An alternative method for measuring present-day rates of relative sea-level rise*

In order to accurately measure present-day RSLR in LECZs, we propose an alternative
approach that combines measurements of shallow subsidence from RSET-MHs with
measurements of deep subsidence and the oceanic component of sea-level rise from GNSS and
satellite altimetry data, respectively (Fig. 5). This approach results in RSLR measurements
expressed with respect to the land surface and eliminates the need for tide-gauge data.
Nevertheless, we stress that best scientific practices will make use of all available data and
compare the results of various measurement techniques. Furthermore, tide gauges remain critical
for measuring many other processes, including tides (the original purpose of tide gauges) and
event-scale phenomena such as storm surge, and remain invaluable in this regard.
In principle, both GNSS stations and tide gauges could be used to measure deep
subsidence and these data could then be combined with measurements of shallow subsidence
(plus geocentric sea-level rise, in the case of GNSS data) to calculate RSLR. However, tide
gauges must have sufficiently long time series (at least 30 years) and known foundation depths to
be useful in this context. In coastal Louisiana, the number of tide gauges that meet these criteria
($n = 5$) are fewer than the number of GNSS stations with known foundation depths ($n = 10$).
Additionally, concerted efforts are currently underway to address the complexities regarding
GNSS monumentation. At a newly constructed subsidence superstation located in the lower
Mississippi Delta, for example, three GNSS instruments are anchored at different depths in order
to obtain a depth-integrated subsidence profile (Allison et al., 2016). Although this type of
analysis is new, it can greatly improve our understanding of subsidence in LECZs in the future.
Furthermore, GNSS data are less susceptible to short-term environmental conditions (i.e. wind
speed and direction, tides, atmospheric pressure changes) than are tide gauge data. Thus, GNSS
is the preferred method for measuring deep subsidence.
Although RSET-MHs, GNSS, and satellite altimetry all have unique limitations,
technology is rapidly improving and reducing these shortcomings. Until recently, for example,
satellite altimetry was ineffective in coastal areas (Cipollini et al., 2017). However, the launch of
the Surface Water and Ocean Topography (SWOT; https://swot.jpl.nasa.gov/home.htm) mission
in 2021 is one of several efforts that are expected to significantly improve the quality of sea-
surface records in the coastal zone and could therefore become an important element of the
approach advocated here (Vignudelli et al., 2011). One remaining limitation of our proposed
method of measuring RSLR is that RSET-MHs are only useful in wetland environments such as
marshes (e.g. Day et al., 2011) and mangroves (e.g. Lovelock et al., 2015). However, space-
based geodetic methods such as interferometric synthetic-aperture radar (InSAR) are effective at
measuring subsidence rates (the sum of shallow and deep subsidence rates) in heavily human-
modified delta environments (e.g. urban areas, agricultural land; Dixon et al., 2006; Jones et al.,
2016; Da Lio et al., 2018), and thus can be complementary to RSET-MH datasets in this context.
Care must be taken though to avoid reliance on permanent scatterers (e.g. buildings) with
foundations at depth that may also not fully capture the shallow subsidence component. Ideally,
RSET-MHs are installed with similar foundation depths as nearby GNSS stations in order to
confirm that the two instruments are neither duplicating nor missing subsidence intervals. In
coastal Louisiana, however, 33% of GNSS stations have no known foundation information, and
this lack of information is likely a common phenomenon worldwide.
Currently, coastal Louisiana has nearly 350 RSET-MHs operated by the USGS as part of
the Coastwide Reference Monitoring System (CRMS; https://lacoast.gov/crms2), which provide
shallow subsidence data at high spatial resolution. Although data from a single RSET-MH are
commonly too noisy to produce a reliable trend (Jankowski et al., 2017), partly because most
RSET-MHs were installed within the last decade and thus have time series that are mostly <10
years long, such a high density of RSET-MHs is not necessary to produce adequate estimates of
shallow subsidence rates for a wider region. Using a Monte Carlo approach, we took random
samples from subsets of the full RSET-MH dataset for coastal Louisiana ($n = 274$) to determine
the smallest sample size that would still produce reasonable outcomes with an acceptable error.
While determining the acceptable error is inherently somewhat arbitrary, the results show that in
coastal Louisiana a minimum of 40 RSET-MHs would be needed in order to produce a mean
shallow subsidence rate with a sufficiently narrow 95% confidence interval (4.54–9.18 mm yr$^{-1}$;
Fig. 6). In terms of density and given the size of coastal Louisiana (25,000-30,000 km$^2$), we
estimate that two RSET-MHs per 1000 km$^2$ would suffice. Although this density is slightly
higher than strictly needed in coastal Louisiana, it is conceivable that higher densities may be
necessary in smaller LECZs.
In addition, averaging data from at least 40 RSET-MHs will encompass the high spatial
variability commonly seen in shallow subsidence. In coastal Louisiana, spatial correlation in
subsidence rates is largely limited to distances <5 km, and no correlation exists beyond 25 km
(Nienhuis et al., 2017). As a result, the relevance of a single measurement of shallow subsidence
is limited to the area immediately around the instrument. Around the world, tide gauges are
generally spaced tens if not hundreds of kilometers apart. Even if tide gauges had benchmarks
anchored at the land surface and were able to measure shallow subsidence, there simply are not
enough tide gauges with records that are sufficiently long for RSLR analysis to capture the large
spatial variability in shallow subsidence. In LECZs worldwide, our ability to predict local rates
of RSLR will improve as more RSET-MHs are added to a growing global network. We therefore
echo Webb et al. (2013) who first proposed this type of global RSET-MH network, arguing that
the instruments are low-cost and produce highly valuable measurements of shallow subsidence.
6.  CONCLUSIONS

In the Mississippi Delta and Chenier Plain of coastal Louisiana, tide gauge benchmarks
and GNSS stations are anchored an average of 21.5 ± 7.4 m and >14.3 ± 11.9 m below the land
surface, respectively. By comparison, the local depth to the Pleistocene surface averages 47 ± 34
m at tide gauge locations and 39 ± 20 m at GNSS stations. Instruments located in the Chenier
Plain, a thin LECZ with Holocene strata typically only 5-10 m thick, are generally anchored in
consolidated Pleistocene strata. In the Mississippi Delta, a LECZ where the Holocene sediment
package is an order of magnitude thicker, tide gauge benchmarks and GNSS stations are
typically anchored within unconsolidated Holocene strata and therefore produce time series that

are very difficult to interpret. Instruments anchored at depth do not capture shallow subsidence, a major component of total subsidence in this area. As a result, tide gauges and GNSS stations in coastal Louisiana, and likely in LECZs worldwide, underestimate rates of RSLR and subsidence with respect to the land surface by a variable but unknown amount.

In order to accurately measure present-day RSLR in LECZs, we propose an alternative method which combines measurements of shallow subsidence from RSET-MHs with measurements of deep subsidence and the oceanic component of sea-level rise from GNSS stations and satellite altimetry, respectively. This approach produces rates of RSLR that are explicitly tied to the land surface and eliminates the need for tide-gauge data in this context. We find that for an area the size of coastal Louisiana, a minimum density of two RSET-MHs per $1000 \text{ km}^2$ is necessary in order to obtain robust shallow subsidence data. We support the call for a global network of RSET-MHs as first put forward by Webb et al. (2013) and recently echoed by Osland et al. (2017). Data from such a global network will help refine existing plans for coastal adaptation that presently may be inadequate to deal with potentially higher-than-anticipated rates of RSLR.

## 7. ACKNOWLEDGEMENTS

This work was supported by the U.S. National Science Foundation (EAR-1349311). We would like to thank Carl Swanson for writing the Python code to run the Monte Carlo analysis, William Veatch for locating benchmark information for USACE tide gauges, and Rena Hoogland (Rijkswaterstaat, The Netherlands) and Marc Hijma (Deltares, The Netherlands) for providing Dutch benchmark data. We appreciate comments on the manuscript provided by Don Cahoon. Thoughtful reviews by Phil Woodworth and an anonymous referee led to considerable improvements.

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

FIGURES

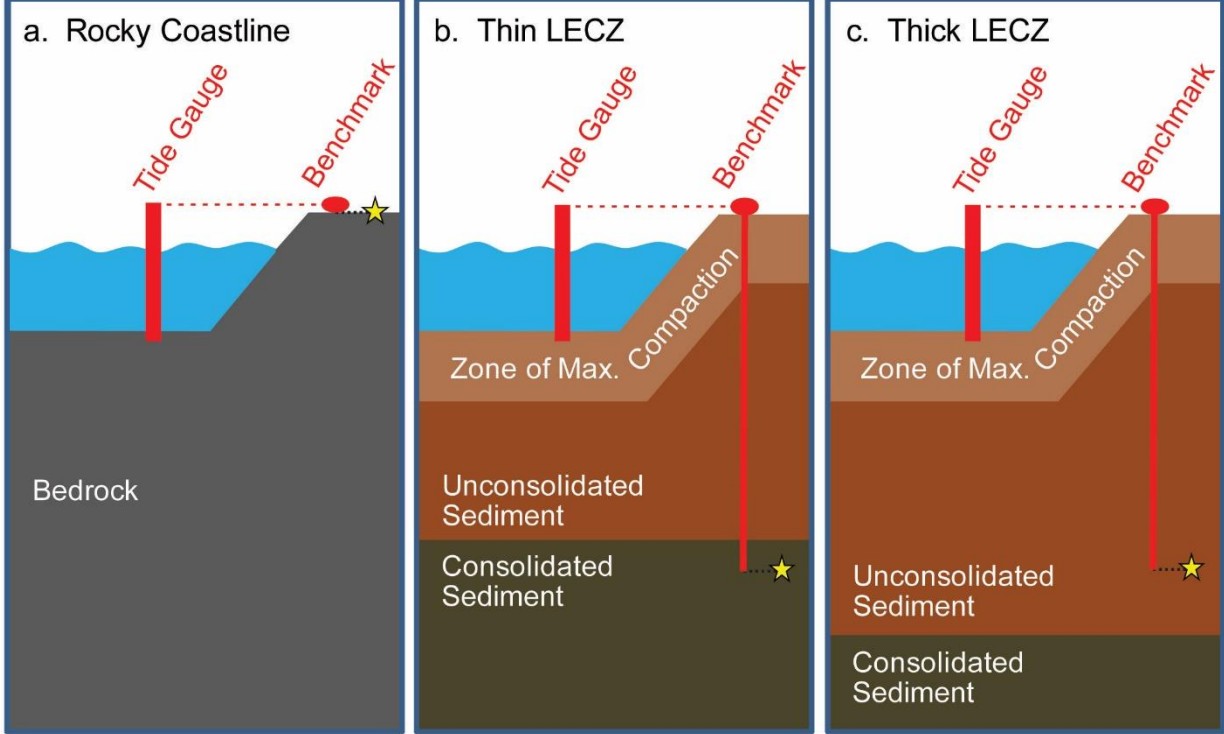

**Figure 1:** Schematic of a tide gauge and associated benchmark on a rocky coastline (a), a thin LECZ (b), and a thick LECZ (c). In all three environments, the tide gauge measures RSLR with respect to the base of the benchmark foundation, which is indicated by a star in each panel.

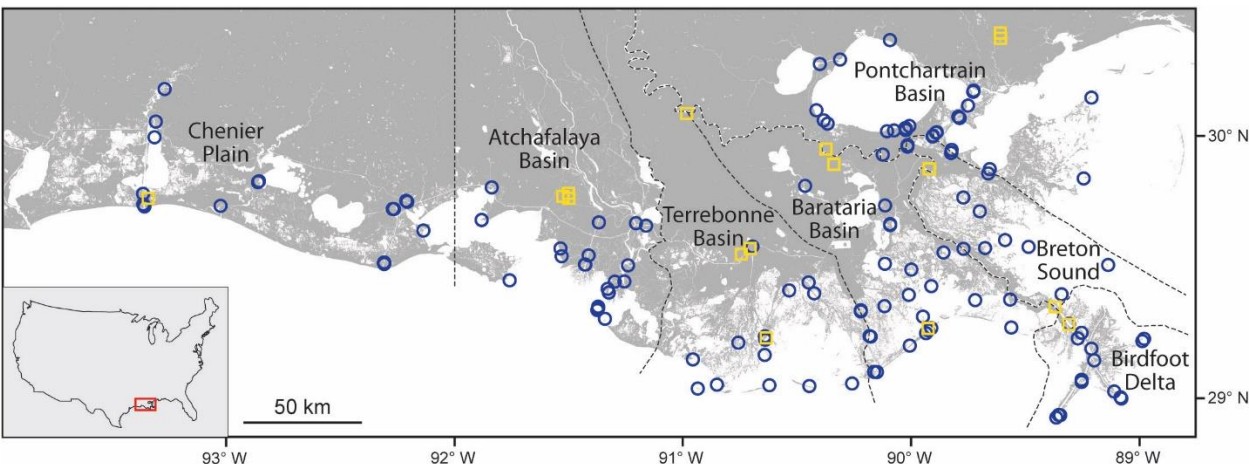

**Figure 2:** Location of tide gauges (circles, $n = 131$) and GNSS stations (squares, $n = 17$) in the Holocene landscape of coastal Louisiana. Dashed lines delineate geographic areas discussed in the text.

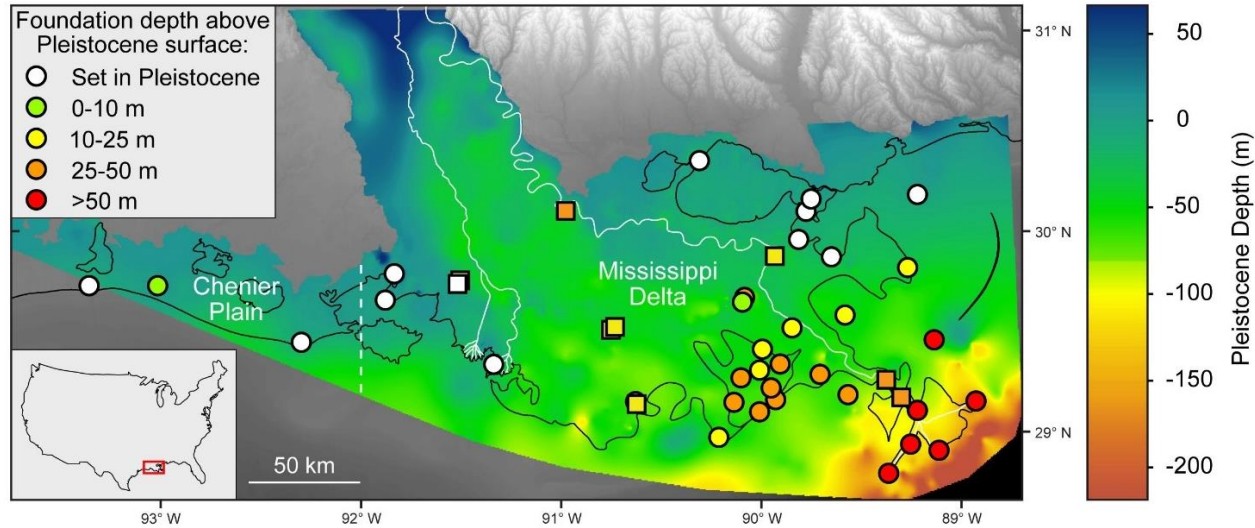

**Figure 3:** Elevation of the Pleistocene surface in coastal Louisiana (with respect to NAVD 88), which approximates
the depth of the Pleistocene surface beneath the land surface given land surface elevations close to mean sea level.
Circles and squares indicate tide gauge and GNSS station locations, respectively, and are color coded according to
foundation height above the Pleistocene surface. Note that two GNSS stations (ENG1 and ENG2, see Table 2) have
the same coordinates (and the same foundation depth) and plot on top of one another. The dashed white line, located
at longitude 92° W, divides the Mississippi Delta from the Chenier Plain. Solid white lines show the Mississippi and
Atchafalaya Rivers. Black lines indicate shorelines. Pleistocene depth information is from Heinrich et al. (2015).

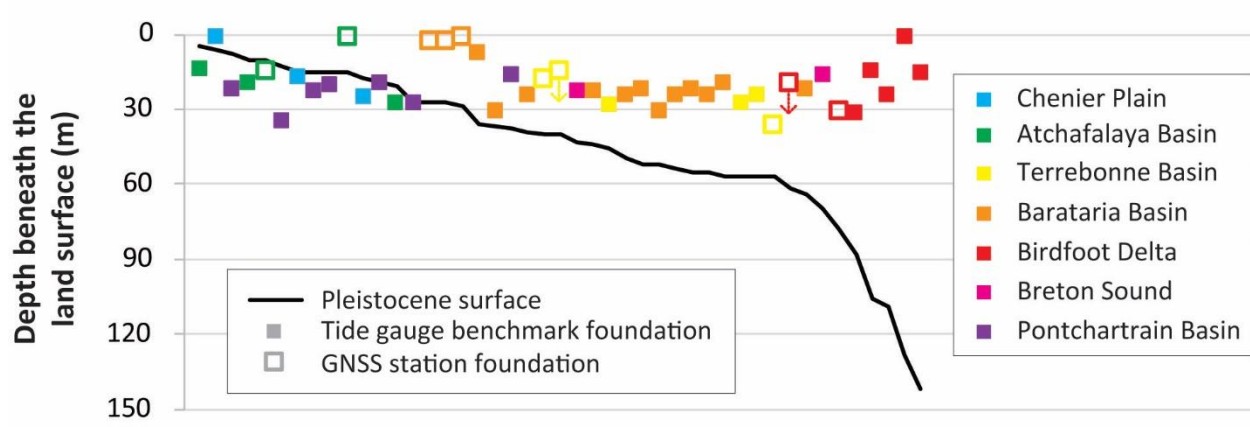

**Figure 4:** Schematic dip-oriented cross section comparing the depth of tide gauge benchmarks and GNSS station
foundations to the local depth to the Pleistocene surface. Sites are arranged by increasing depth of the Pleistocene
surface. Note that two GNSS stations have minimum foundation depths (see Table 2), indicated here by small,
downward-pointing arrows. See Figure 2 for the location of geographic areas.

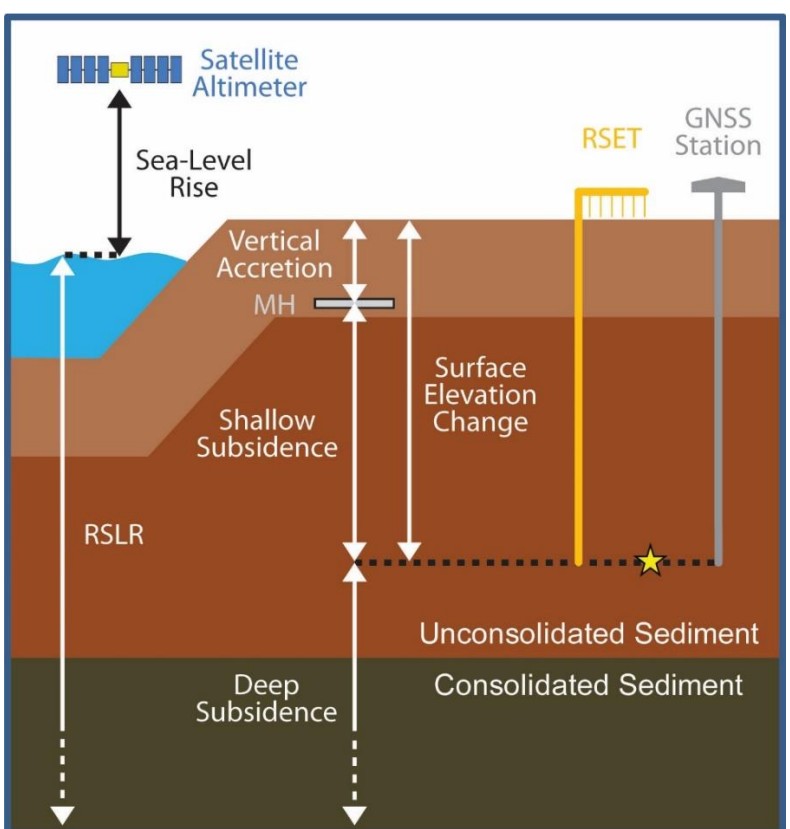

**Figure 5**. Schematic of combined instrumentation that includes a RSET-MH, which measures shallow subsidence,
and a GNSS station, which measures deep subsidence. To measure shallow subsidence using a RSET-MH, surface
elevation change is subtracted from vertical accretion (Cahoon, 2015). Surface elevation change is the change in
height from a horizontal arm at a fixed elevation to the wetland surface, measured using vertical pins. Vertical
accretion is the thickness of sediment that accumulates above a feldspar marker horizon. If constructed with similar
foundation depths (as shown by the star), the RSET-MH and GNSS station collect data that are complementary and
can be added together and combined with satellite altimetry data to calculate the rate of RSLR.

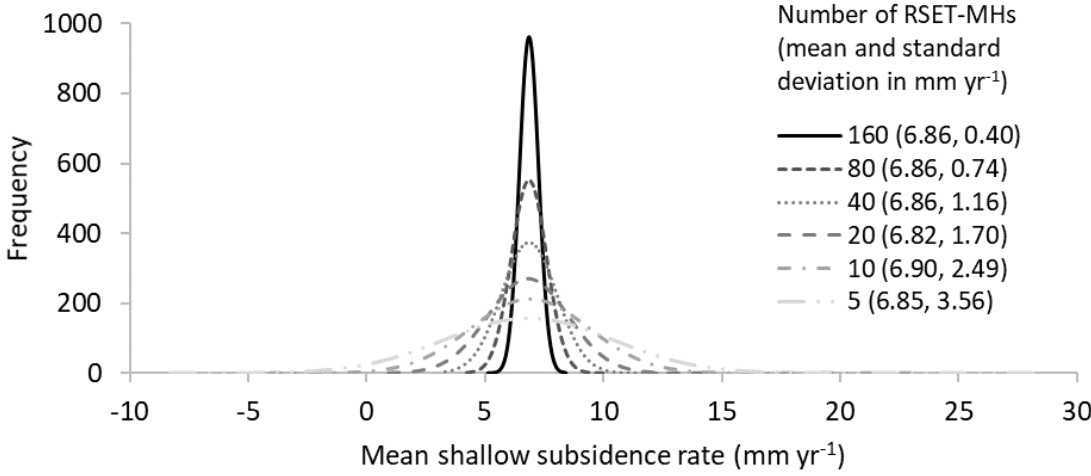

**Figure 6**. Probability density functions of the mean shallow subsidence rate for a given number of RSET-MHs,
calculated using a Monte Carlo simulation and 10,000 randomizations per analysis.
TABLES

**Table 1:** Tide gauges in the Holocene landscape of coastal Louisiana with known foundation information (*n* = 35).

| Tide gauge name | Agency | Latitude | Longitude | Maximum benchmark foundation depth (m) | Depth to Pleistocene surface (m) | Benchmark foundation height above Pleistocene surface (m) |
|---|---|---|---|---|---|---|
| Amerada Pass | NOAA | 29.4500 | -91.3383 | 27.4 | 21 | Set in Pleistocene |
| Barataria Waterway | USACE | 29.6694 | -90.1106 | 7.4 | 36 | 29 |
| Bay Gardene | NOAA | 29.5983 | -89.6183 | 23.2 | 43 | 20 |
| Bay Rambo | NOAA | 29.3617 | -90.1400 | 24.4 | 54 | 30 |
| Bayou Petit Caillou | USACE | 29.2543 | -90.6635 | 24.4 | 57 | 33 |
| Bayou St. Denis | NOAA | 29.4967 | -90.0250 | 23.2 | 44 | 21 |
| Billet Bay | NOAA | 29.3717 | -89.7517 | 21.9 | 52 | 30 |
| Breton Island | NOAA | 29.4933 | -89.1733 | 16.8 | 70 | 53 |
| Calcasieu Pass | NOAA | 29.7683 | -93.3433 | 25 | 18 | Set in Pleistocene |
| Caminada Pass | NOAA | 29.2100 | -90.0400 | 21.9 | 55 | 33 |
| Chef Menteur Pass | NOAA | 30.0650 | -89.8000 | 35.1 | 13 | Set in Pleistocene |
| Comfort Island | NOAA | 29.8233 | -89.2700 | 16.8 | 38 | 21 |
| Cypremort Point | NOAA | 29.7133 | -91.8800 | 19.4 | 10 | Set in Pleistocene |
| East Bay | NOAA | 29.0533 | -89.3050 | 14.6 | 106 | 91 |
| East Timbalier Island | NOAA | 29.0767 | -90.2850 | 28.8 | 46 | 17 |
| Freshwater Canal Locks | NOAA | 29.5517 | -92.3050 | 17.1 | 15 | Set in Pleistocene |
| Grand Isle | NOAA | 29.2633 | -89.9567 | 19.8 | 57 | 37 |
| Grand Pass | NOAA | 30.1267 | -89.2217 | 23.2 | 15 | Set in Pleistocene |
| Greens Ditch | NOAA | 30.1117 | -89.7600 | 21.9 | 8 | Set in Pleistocene |
| Hackberry Bay | NOAA | 29.4017 | -90.0383 | 30.5 | 52 | 22 |
| Lafitte | NOAA | 29.6667 | -90.1117 | 30.5 | 37 | 7 |
| Lake Judge Perez | NOAA | 29.5583 | -89.8833 | 24.4 | 39 | 15 |
| Leeville | NOAA | 29.2483 | -90.2117 | 28 | 57 | 29 |
| Martello Castle | NOAA | 29.9450 | -89.8350 | 19.51 | 19 | Set in Pleistocene |
| Mendicant Island | NOAA | 29.3183 | -89.9800 | 24.4 | 55 | 31 |
| Mermentau River | USACE | 29.7704 | -93.0135 | 1.5 | 6 | 5 |
| North Pass | NOAA | 29.2050 | -89.0367 | 15.2 | 142 | 127 |
| Pass Manchac | NOAA | 30.2967 | -90.3117 | 20.7 | 15 | Set in Pleistocene |
| Pelican Island | NOAA | 29.2667 | -89.5983 | 21.9 | 64 | 42 |
| Pilottown | NOAA | 29.1783 | -89.2583 | 32 | 88 | 56 |
| Port Eads | USACE | 29.0147 | -89.1658 | 0.9 | 128 | 127 |
| Shell Beach | NOAA | 29.8683 | -89.6733 | 27.4 | 27 | Set in Pleistocene |
| Southwest Pass | NOAA | 28.9250 | -89.4183 | 24.4 | 109 | 85 |
| St. Mary's Point | NOAA | 29.4317 | -89.9383 | 24.4 | 50 | 26 |
| Weeks Bay | NOAA | 29.8367 | -91.8367 | 14.3 | 5 | Set in Pleistocene |


**Table 2:** GNSS stations in the Holocene landscape of coastal Louisiana with known foundation information (*n* =
716 10).

| GNSS station code | Latitude | Longitude | Foundation depth (m) | Depth to Pleistocene surface (m) | Foundation height above Pleistocene surface (m) | Data source |
|---|---|---|---|---|---|---|
| AWES | 30.10 | -90.98 | 1 | 29 | 28 | Karegar et al. (2015) |
| BVHS | 29.34 | -89.41 | >20 | 62 | <42 | Dokka et al. (2006); Karegar et al. (2015) |
| ENG1 | 29.88 | -89.94 | ~3 | 27 | ~24 | Karegar et al. (2015) |
| ENG2 | 29.88 | -89.94 | ~3 | 27 | ~24 | Dokka et al. (2006) |
| FRAN | 29.80 | -91.53 | 14.7 | 10 | Set in Pleistocene | Dokka et al. (2006) |
| FSHS | 29.81 | -91.50 | 1 | 15 | 14 | Karegar et al. (2015) |
| HOMA | 29.57 | -90.76 | 18.3 | 40 | 22 | Dokka et al. (2006) |
| HOUM | 29.59 | -90.72 | >15 | 40 | <25 | Dokka et al. (2006); Karegar et al. (2015) |
| LMCN | 29.25 | -90.66 | 36.5 | 57 | 21 | Dokka et al. (2006); Karegar et al. (2015) |
| VENI | 29.28 | -89.36 | 30.5 | 78 | 48 | Dokka et al. (2006) |

**Table 3.** Holocene sediment thicknesses of LECZs around the world, measured close to the shoreline where coastal
strata tend to be the thickest.

| Low-elevation coastal zone | Maximum thickness (m) | LECZ type | Reference |
|---|---|---|---|
| Chenier Plain, Miranda, New Zealand | 3-5 | thin | Woodroffe et al. (1983) |
| Chenier Plain, SW Louisiana, USA | 5-10 | thin | Heinrich et al. (2015) |
| Venice Lagoon, Italy | 10-15 | thin | Zecchin et al. (2009) |
| Chao Phraya Delta, Thailand | 10-15 | thin | Tanabe et al. (2003a) |
| Vistula Delta, Poland | 10-20 | thin | Mojski (1995) |
| Rhine-Meuse Delta, The Netherlands | 20-25 | thick | Hijma et al. (2009) |
| Huanghe Delta (modern), China | 20-25 | thick | Xue (1993); Yi et al. (2003) |
| Po Delta, Italy | 20-25 | thick | Amorosi et al. (2017) |
| Tokyo Lowland, Japan | 20-60 | thick | Tanabe et al. (2015) |
| Mekong Delta, Vietnam | 25-40 | thick | Ta et al. (2002); Tanabe et al. (2003b) |
| Nobi Plain, Japan | 30-40 | thick | Hori et al. (2011) |
| Shatt al-Arab Delta, Iraq | 30-40 | thick | Larsen (1975) |
| Nile Delta, Egypt | 30-50 | thick | Stanley and Warne (1993) |
| Song Hong Delta, Vietnam | 35-40 | thick | Funabiki et al. (2007) |
| Fly Delta, Papua New Guinea | 35-45 | thick | Harris et al. (1993) |
| Ganges-Brahmaputra Delta, Bangladesh | 50-100 | thick | Goodbred and Kuehl (2000) |
| Mississippi Delta, SE Louisiana, USA | 50-100 | thick | Heinrich et al. (2015) |
| Yangtze Delta, China | 60-90 | thick | Li et al. (2000) |
| Indus Delta, Pakistan | 110-120 | thick | Clift et al. (2010) |


**Table 4.** Benchmark foundation depths and local depth to the Pleistocene surface for tide gauges in The
Netherlands. Benchmark depths from R. Hoogland (personal communication, 2018). Pleistocene surface depths are
from Vos et al. (2011).

| Tide gauge name | Agency | Latitude | Longitude | Benchmark foundation depth (m) | Depth to Pleistocene surface (m) | Benchmark foundation height above Pleistocene surface (m) |
|---|---|---|---|---|---|---|
| Vlissingen | Rijkswaterstaat | 51.4422 | 3.5961 | 17.6 | 4-6 | Set in Pleistocene |
| Hoek van Holland | Rijkswaterstaat | 51.9775 | 4.1200 | 14 | 20-22 | 6-8 |
| IJmuiden | Rijkswaterstaat | 52.4622 | 4.5547 | 13 | 18-20 | 5-7 |
| Den Helder | Rijkswaterstaat | 52.9644 | 4.7450 | 5-25 | 2-4 | Set in Pleistocene |
| Harlingen | Rijkswaterstaat | 53.1756 | 5.4094 | 5-25 | 4-6 | Likely set in Pleistocene |
| Delfzijl | Rijkswaterstaat | 53.3264 | 6.9331 | 20 | 6-8 | Set in Pleistocene |
