# Peer review of "Measuring rates of present-day relative sea-level rise in low-elevation coastal zones: A critical evaluation Molly E. Keogh1 and Torbjörn E. Törnqvist Department of Earth and Environmental Sciences, Tulane University, 6823 St. Charles Avenue, New Orleans, Louisiana 70118-5698, USA 1<"

_Ocean Science, 2018_

## Referee Comment (RC1) · P.L. Woodworth (Referee) · 25 Aug 2018

25 August 2018

Comments on "Measuring rates of present-day relative sea-level rise in low-elevation coastal zones: A critical evaluation" by Keogh and Tornqvist (OSD)

This paper makes use of a data set of benchmark (BM) depths at tide gauges and GPS stations in Louisiana, which enables the authors to come to conclusions regarding the ability of tide gauges to make accurate measurements of relative sea level rise in this and similar deltas. They make some recommendations on how such measurements

might be done better.

This is short paper which is mostly written well with decent figures. I am sure that the topics addressed have been discussed by these and other authors previously. Also they do not produce any actual new results on relative sea level trends in the area. Nevertheless the BM data set does result in a nice couple of plots which enable them to make their main point well. So I would have no objection to seeing this paper published eventually, although I do have some comments on their arguments and on the way some of the text is written.

One comment is a technical issue to do with the way that NOAA works. The authors say correctly that there are typically half a dozen BMs at each tide gauge site. Many of these are deep ones and Table 1 lists the depths of the deepest in each case. If the datum of the tide gauges is defined relative to one of these deep marks, then I can understand the arguments of the authors that relative sea level rise could be underestimated.

However, sometimes there are also surface (or near surface) marks which can appear as 'zero depth (N/A setting)' in Table S1 of the paper. Now, the NOAA web site (https://tidesandcurrents.noaa.gov/datum_options.html#STND) states explicitly that:

"Station datum is referenced to the primary bench mark at the station for the definition of the tide gauge station datum".

So, if the designated primary mark is a surface mark, the Station Datum at the gauge will have been defined by the land surface and their arguments will not apply.

Now, the only important site in the delta with a decent long record is Grand Isle. That has data from 1947 and its benchmark sheet (available from the NOAA web site) shows that the primary mark is BM10 which is a "survey disk on the sea wall" (again shown as zero depth and N/A setting in Table S1). This is a surface mark so the arguments of the authors do not apply here.

I looked at the information on the NOAA web site for all 31 NOAA stations given in Table 1 of the paper (i.e. the 35 stations listed minus 4 USACE stations). The NOAA web site information is essentially the same as in Table S1. Of the 31, 6 have primary marks which are surface (or very near surface) marks: Caminada Pass, East Bay, Grand Isle, Lafitte, Martello Castle and Weeks Bay. If the authors agree with this then I think their text should mention it.

Just in case, I checked my interpretation about the way NOAA works with the CO-OPS Technical Director (Dr. Peter Stone) and Chief Scientist (Dr. Greg Dusek). They replied:

"We control the water level observation primarily off of one primary bench mark (PBM) and then ensure the stability of that mark by using the remaining 9 or so marks. On occasions when we see substantial and/or continual differential movement between the PBM and the other marks, we adjust the PBM to a different mark determined to be stable relative to the remaining marks."

So that confirms what is on the NOAA web site, and confirms that my comments about the six mentioned above, and Grand Isle in particular, are correct. They do not fit into the main argument of the paper, so there should be some extra wording to handle that.

As for the other 25 stations in Table 1 for which the primary mark is a deep one, then I agree with their comments, but only in principle, and only at a time way into the future when these stations will have acquired records long enough for trend estimation. Stone and Dusek remarked:

"The large number of tide gauges used in the analysis is very perplexing. The NOAA gauges [mentioned in Table 1] (which were installed by CO-OPS) were installed for wide ranges of time. Two of the gauges (Shell Beach and Grand Isle) were installed for decades and we have calculated relative sea level change rates. The others have only been installed for a few months or years and do not have enough data to calculate statistically significant RSLR [relative sea level rise]."

Now, Grand Isle I have already mentioned. In fact, Shell Beach has a deep primary mark, so I accept that the argument of the authors applies for that. But as Shell Beach has data (in the PSMSL) only for 2008-2017, it is hardly yet a long record.

So I think some care should be taken in the text between explaining what could happen IN PRINCIPLE regarding tide gauges with deep primary marks, and what is the real situation at the moment in the delta.

This takes me to two mentions of the PSMSL in the paper. At line 50 the authors state that there are 5 PSMSL stations in Lousiana but do not give their names. They are Eugene Is (data 1939-1974), Grand Isle (1947-2017), South Pass (1980-1999), Shell Beach (2008-2017) and New Canal Station (2006-2017). As mentioned above, Grand Isle is the only important one for sea level trends. The PSMSL defines RLR datum at Grand Isle (and other NOAA sites) using the Station Datum information in each case that NOAA provides. Therefore, I think there should be a mention somewhere in the paper to the effect that the sea level rate at Grand Isle provided by the PSMSL record is not likely under-estimated as the text presently implies.

The other mention of the PSMSL is in the paragraph at lines 250-261. It again mentions only 5 PSMSL stations in the area. Why? The PSMSL cannot be expected to databank the density of stations that the authors need, so to somehow conflate the PSMSL with that requirement seems strange to me. In fact, what the PSMSL would be happy with in an area this size is a single tide gauge station with GPS and good BM control. Anyway, the authors show potentially they have many more than 5 so what is their problem? Also the paragraph says that of the 5 'only a few' have RSET nearby. If one takes 'a few' as meaning 3 or similar then one could read this sentence as saying that most PSMSL stations have RSET, which I think is opposite to what the authors want to say! So this paragraph needs rewording and I can't see why the PSMSL is being dragged into it at all.

Conclusions - so I see the problems that the authors raise about deep BMs, in principle.

However, I do not buy the suggestion that, instead of tide gauges, a better job could be done using RSET-MH data which seems to me to be a very rough and ready method, combined with GPS for deep submergence, combined with altimetry. RSET, GPS and altimetry data all have their own nuances and problems, and in particular altimetry until fairly recently has had problems getting very close to the coast.

Tide gauges could do the job you want if you have at least one surface mark at each site, and if there is ongoing monitoring of the relative heights between surface and deep BMs. That would solve your problem; a conversation with NOAA is required about constructing a history of the evolution of relative heights between benchmarks at each site.

Stone and Dusek commented to me also that "The authors' did not address the mounting of the water level sensors on different structures they were comparing and how those structures can be affected by settling. Some of the stations used in their comparisons are probably installed on piers where the pilings may only be sunk a few feet. Others, like the water level stations at Shell Beach and Calcasieu Pass, LA are mounted on massive steel structures driven into consolidated sediments (which we refer to as SPIPs). The type of installation can be relevant to consider when attempting to accurately assess sensor movement relative to bench marks on land, and presumably in the cases of the SPIPs, our leveling data could indicate if deep rod marks and the shallower rod or concrete marks show variable long-term trends relative to the water level observations."

So, while accepting the general main point of the authors, I think the main thing is to have access to histories of all the relevant surveying information at a site.

A last comment about the Conclusions is an obvious one, that the correct scientific approach is to make use of data from all techniques and see eventually how they compare, not just suggest rejecting tide gauges (which NOAA pay for, given that they are anyway needed for monitoring transient events such as storm surges) by adopting

an 'alternative approach'.

The Conclusions also makes some comments about deltas elsewhere around the world and lists some in Table 3. How many have deep BMs like in Louisiana? I suspect most do not, but at best have surface marks in which these arguments will not apply. It would be interesting to know.

So for the reasons above I think some rewriting of the text is required.

Detailed comments:

line 17 and elsewhere - GPS is better denoted at GNSS (Global Navigation Satellite System) these days.

line 35 - a reference to long tide gauge records in N Europe and N America could be Woodworth et al. (Surveys in Geophysics, 2011). The longest US record was claimed for many years to be Key West (Maul and Martin, GRL, 1993) but I guess now one should also mention Boston (Talke, JGR, 2018).

line 49 - the PSMSL should be referenced by its web site and journal (http://www.psmsl.org and Holgate et al., J Coastal Res, 2013)

line 50 - give the names of the five (see above)

line 89 - these references should also include the IOC Manuals, see http://www.psmsl.org/train_and_info/training/manuals/

line 125 - a reference is needed for where you got the Pleistocene surface information from.

line 169 - 'because all tide gauge benchmarks'. This is not true, see above.

line 172 - I would be grateful if you did not use the word 'eustatic' which means different things to different people (there is a recommendation about this in one of the IPCC reports). I suggest this is reworded:

... deep subsidence plus the component of RSLR associated with changes in real ocean level.

(or something like that). And drop 'as well .... effects'.

line 186 - I would reword:

... includes subsidence of that part of ... sediments deeper than the BM depth.

line 207 - reword to avoid eustatic:

.. adding the historic rate of real (geocentric) sea-level rise ..

There is a reference to Ericson et al. (2006) in the context of not using tide-gauge data. But the sea level rise value in that paper was just the global average taken from the IPCC (1.5 mm/yr) which hardly seems to me to be superior to using local tide gauges where available. I realise why Ericson et al. had to do that in their paper but it is not to be recommended in your case.

line 223 - a reference is needed for the InSAR mention, preferably for its use in deltas.

230-233 - SWOT is only one of several efforts to improve coastal altimetry. A general reference, in which there is mention of SWOT, would be:

Vignudelli, S., Kostianoy, A., Cipollini, P and Benveniste, J. (eds). 2011. Coastal altimetry. Berlin: Springer Publishing. 578pp.

237 - 'commonly too noisy'. From what I have read of the method I'm not surprised!

References - need doi's adding

453 - there should be an accent over the 'i' in Miguez

Figure 1 - What are the short and long vertical lines beneath the tide gauge in each panel supposed to be showing. I'd remove them. The point is that the datum of a tide gauge is determined by levelling to the BM nearby, so I would have the horizontal red line for the tide gauge at the same level as the BM and a dotted line between them.

I think this figure may have been adapted from Figure 1 of Webb et al. (2013) which in their case has a short vertical band which I think is supposed to be indicating a float gauge (of which there are fewer around), and a longer vertical band which I think has the same function as the vertical red line for the BM in the present case. Anyhow, please lose the vertical lines under the tide gauges in this case.

Figure 2 and 3 - could the lat/lon ticks and annotation face outside the map to be clearer?

Figure 3 - could the colour scale on the right be labelled Pleistocene Depth and the insert headed FD above PS or similar? As mentioned above, a reference is needed in the caption for the Pleistocene depth information. The black lines for the shoreline are hard to see.

There is no point mentioning ENG1 and 2 if they don't appear on the plot. But perhaps say (ENG1 andd ENG2, see Table 2).

Figure 4 - nice plot.

Figure 6 - put mm/yr after mean, standard deviation

Tables 1 and 2 - head the column 'Maximum benchmark foundation depth (m)'

Figure S1 - I don't see the point of this figure. It has no more information than Figure 6. Doesn't do any harm I guess.

I hope these comments are useful. I have no objection to my identity being revealed. I am very grateful to Peter Stone and Greg Dusek for information which helped me complete this review.

---

## Referee Comment (RC2) · Anonymous Referee #2 · 20 Sep 2018

**General comments:**

This study seems closely related to the work published in GSA Today (Nienhuis et al. 2017) in which the authors were involved. But it is not clearly stated how both relate together. Nienhuis et al. is quoted towards the end of the manuscript, just before the conclusions. The findings on the underestimation due to shallow subsidence are already present in Nienhuis et al. Hence, should this manuscript be considered as supplemental information to the GSA Today one? I think it is important that the authors clarify how both studies relate together from the beginning (introduction). In addition, the introduction and the conclusion ("we present", "we propose") suggest the approach

is novel. However, later on we find expressions and references which suggest it is not. Overall the authors need to make an effort to unambiguously set their study in the scientific context.

In my opinion, the manuscript suffers from a perspective bias "against" tide gauges. That is, the authors show that both the tide gauges and GPS antennas are similarly anchored deep below the surface (at almost equivalent depths). Thus, none of them can actually capture the shallow subsidence. The combination of satellite altimetry and GPS data or the use of tide gauges suffer from the same drawback. Consequently, the statement that the novel approach eliminates the need for tide gauge data (repeated several times in the manuscript) is not objectively supported, because the same criticism applies to GPS antennas, and hence to the combination of satellite altimetry and GPS data. From my understanding, tide gauges + RSET-MH can work as well as satellite altimetry + GPS + RSET-MH. The authors need to think about it, and provide arguments to support their claim in a more convincing way, or reconsider the presentation of their findings (which are anyway interesting, in my opinion).

The manuscript (introduction) suggests an assessment of their findings in LECZs worldwide, but the authors do not provide evidence that the findings apply beyond their case study zone, except for some general considerations (sediment thick in different coastal areas of world from the literature). The authors should be aware that different countries (agencies) have different practices in building infrastructures (tide gauges or GPS antenna monumentations). The US case study is likely not representative of the wide range of practices elsewhere. They should consider reducing the scope of the claims, and develop a cautious discussion in extending the findings in LECZ worlwide. The title may be revisited too.

In line with the above comment, I would suggest a search in the literature about GPS station monumnetations to support the worldwide extension. I vaguely remember a talk a decade ago or so about GPS antenna monumentations within an IGS meeting or an IAG scientific assembly. The concern of the study was the ability of the different types
of GPS antenna monumentations to estimate actual ground / crustal motions. I think it might be worth searching for the details of this study or later studies on this subject.

In addition to the above comment, the choice of the deepest benchmark in section 4 needs to be supported, especially regarding the leveling analysis and practice to maintain the tide gauge datum, which can differ from one country (agency) to another (agency). Furthermore, I think this methodological choice should not be presented / discussed in the "Results" section but in the methods section.

The manuscript is overall well written with good illustrations (Figures). In my opinion, it needs to consider the above comments. My suggestion is therefore a major revision.

Specific comments & Technical corrections:

p.2, L17-18: The expression is confusing. That is, if the station is >14 m, it includes the surface, and thus can capture any land motion. Consider rephrasing, why not using the same form as with the tide gauges? (Simply remove ">").

p.2. L22: the need for tide gauge data is often multi-application. The authors should be careful with this claim, and state the context of it (eliminates the need for this specific application and LECZ situation). In addition, see major comment above, that is, the same concerns apply to the GPS monumnetation, hence both tide gauges and GPS show the same drawback.

p.2, L34: a reference to support this claim is missing. It could be Holgate et al. (2013) which describes a data bank or similar; it could be an (the) article(s) that rescued the historical data of the stations listed in brackets.

p.2. L37: Watson et al. is a good paper but it is not relevant in the context of this sentence. (Its global sea-level rise estimate is based on satellite altimetry data). Maybe the reference can be used somewhere else.

p.3 L50. Consider adding the reference for the PSMSL (Holgate et al. 2013 in J. Coastal Res).

OSD
p.3. L53. What signals encompass "natural variability" here?

p.4, L93. A reference is needed to support this claim. I vaguely remember a talk several years (decade?) ago at an IGS or IAG meeting about GPS antenna monumentations (structure, depth...) with some statistics. The concern of the study was the ability of GPS antennas to estimate actual ground / crustal motions. I think it can be worth searching the literature, especially since L97 states the issue of the nature of GPS station foundations as an objective of the study.

p.4. L100. Confusing (see general comments above). The expression suggests the approach is novel, especially because in the previous sentence it is stated what is not the purpose of the study. However, there are two references at the end of the sentence. Is this study a refinement? Consider rephrasing and clarifying.

p.5, L144-145. This choice needs to be supported, especially regarding the leveling analysis and practice to maintain the tide gauge datum, which can differ from one country (agency) to another (agency).

p.6, L172 (L205 too). What is behind the term 'eustatic'?

p.14. I cannot see whether there are squares and circles co-located. Consider using a different colour too, it may help.

OSD

---

## Author Response (AR1)

**Author Response**

This document includes a list of key changes made to the manuscript, a point-by-point response to the two reviews, and a marked-up manuscript version.

Key manuscript changes:

- Added an analysis of mean foundation depth of primary benchmarks for comparison with our original analysis of deepest known benchmarks. We find that primary benchmark foundation depths are indistinguishable from the dataset as a whole.

- Added further analysis of data presented in Jankowski et al. (2017), suggesting that shallow subsidence occurs dominantly in the uppermost 5 m of wetland stratigraphy.

- Added data on the foundation depths of tide gauge benchmarks in the Netherlands, which support our assertion that the issues discussed in this manuscript are likely global in scope.

- Expanded our discussion to include benchmarks mounted on concrete structures with unknown foundation depths; we suggest that these structures are likely anchored at some depth below the surface and thus continue to support the main argument of our paper: that tide gauges with benchmarks anchored at depth do not record all shallow subsidence.

- Clarified that tide gauges remain critical for measuring many processes (e.g. tides, storm surge) and that we are merely discussing a specific (yet important) context where tide-gauge data may not be the best option.

- Expanded discussion of the limitations of various instruments and methods of measuring RSLR and further describe data analysis techniques that could be used to overcome some of these shortcomings.

- Clarified the novelty of work presented here and improved description of how it fits into the recent literature.

Response to the manuscript review by Philip Woodworth
Reviewer comments are in bold text; author responses are in normal text
* * *
**This paper makes use of a data set of benchmark (BM) depths at tide gauges and GPS stations in Louisiana, which enables the authors to come to conclusions regarding the ability of tide gauges to make accurate measurements of relative sea level rise in this and similar deltas. They make some recommendations on how such measurements might be done better.**

We greatly appreciate this review by Dr. Philip Woodworth and his expertise in tide gauge data analysis. We also value the input from the NOAA colleagues. We have closely followed the recommendations provided in this review and believe that our manuscript is considerably improved as a result.

**This is short paper which is mostly written well with decent figures. I am sure that the topics addressed have been discussed by these and other authors previously. Also they do not produce any actual new results on relative sea level trends in the area. Nevertheless the BM data set does result in a nice couple of plots which enable them to make their main point well. So I would have no objection to seeing this paper published eventually, although I do have some comments on their arguments and on the way some of the text is written.**

To our knowledge, our study is actually the first systematic investigation of the foundation depths of tide gauge benchmarks and GNSS stations and the resulting implications for measurement of relative sea-level rise (RSLR). Our purpose is not to reinvent the wheel by reanalyzing time series. Instead, we draw attention to a limitation of tide-gauge data and present an alternative approach to measuring RSLR in low-elevation coastal zones.

**One comment is a technical issue to do with the way that NOAA works. The authors say correctly that there are typically half a dozen BMs at each tide gauge site. Many of these are deep ones and Table 1 lists the depths of the deepest in each case. If the datum of the tide gauges is defined relative to one of these deep marks, then I can understand the arguments of the authors that relative sea level rise could be underestimated.**

**However, sometimes there are also surface (or near surface) marks which can appear as 'zero depth (N/A setting)' in Table S1 of the paper. Now, the NOAA web site (https://tidesandcurrents.noaa.gov/datum_options.html#STND) states explicitly that: "Station datum is referenced to the primary bench mark at the station for the definition of the tide gauge station datum".**

Excellent point. We have added the clarification that NOAA tide gauges are typically leveled using a benchmark designated as the primary benchmark; secondary benchmarks are used to assess the stability of the primary benchmark. See lines 95-97.

Please note that no benchmarks listed in Table S1 are anchored at depth = 0. The shallowest benchmark is anchored at 0.9 m. A few benchmarks have unknown foundation depths (which are listed in Table S1 as

"unknown"), but this should not be interpreted to be foundation depth = 0. This clarification has been added to the manuscript in lines 187-188. Several other tide gauges listed in Table S1 have no published benchmarks; in this case, the benchmark setting type and foundation depth are left blank. This is now clarified in the header of Table S1.

**So, if the designated primary mark is a surface mark, the Station Datum at the gauge will have been defined by the land surface and their arguments will not apply.**

In principle, we agree with this assessment: a tide gauge leveled to a primary benchmark that is anchored at the ground surface records RSLR with respect to the ground surface. However, we argue that few, if any, benchmarks in coastal Louisiana are actually anchored at ground level. See below for details. In Table S1, we have added a column listing the depth of primary benchmark foundations, wherever known.

**Now, the only important site in the delta with a decent long record is Grand Isle. That has data from 1947 and its benchmark sheet (available from the NOAA web site) shows that the primary mark is BM10 which is a "survey disk on the sea wall" (again shown as zero depth and N/A setting in Table S1). This is a surface mark so the arguments of the authors do not apply here.**

This is an important point and we have added detailed clarification in the manuscript. Table S1 includes two Grand Isle tide gauges. The older gauge recorded data from 1947-1980; the newer gauge has been operational since 1979. Although the two datasets are typically combined, the tide gauges have different station numbers assigned by NOAA and the older gauge is not associated with any currently-published benchmarks. The newer gauge has 5 benchmarks, and, as noted by the reviewer, the primary one is set on a seawall. Note that Table S1 indicates that this primary benchmark has an unknown foundation depth rather than listing it as foundation depth = 0.

Although the primary Grand Isle benchmark is indeed mounted on a concrete seawall, we disagree that it is a surface mark with foundation depth = 0. Just as a benchmark mounted on a steel rod driven to depth responds to changes in elevation with respect to the base of the rod, a benchmark mounted on a concrete structure responds to elevation changes with respect to the foundation of the concrete structure. Although we were unable to acquire construction details for the seawall, it is highly unlikely that the seawall is simply resting on the ground surface. We expect that the seawall foundation extends at least several meters into the subsurface in order to provide stability and protection to the adjacent Grand Isle Coast Guard station. If this is indeed the case, the primary Grand Isle benchmark should NOT be considered a surface mark. That said, we agree that it is conceivable that the foundation depth of the primary benchmark at Grand Isle is considerably shallower than that of the deepest benchmark (19.8 m). We have added this important point in the text (see lines 235-248), recognizing that this may reduce the underestimation of the rate of RSLR at this tide gauge.

This important comment prompted us to carry out some further analysis of data presented in Jankowski et al. (2017), suggesting that shallow subsidence occurs dominantly in the uppermost 5 m of the wetland stratigraphy in this region. Using data from 274 monitoring stations across coastal Louisiana, Jankowski et al. (2017) calculated a mean shallow subsidence rate of $6.8 \pm 7.9$ mm yr$^{-1}$. Limiting this analysis to stations where the instrument is anchored in Pleistocene strata and the overlying (Holocene) strata are <5

m thick, we find a mean shallow subsidence rate of $6.4 \pm 5.4$ mm yr$^{-1}$. The similarity in these two calculated shallow subsidence rates suggests that even tide gauges with associated benchmarks anchored only a few meters deep do not fully capture the shallow subsidence signal. We have included discussion of this additional analysis in lines 249-257 of the manuscript.

**I looked at the information on the NOAA web site for all 31 NOAA stations given in Table 1 of the paper (i.e. the 35 stations listed minus 4 USACE stations). The NOAA web site information is essentially the same as in Table S1. Of the 31, 6 have primary marks which are surface (or very near surface) marks: Caminada Pass, East Bay, Grand Isle, Lafitte, Martello Castle and Weeks Bay. If the authors agree with this then I think their text should mention it.**

We have added a column to Table S1 that lists the depth of the primary benchmark foundation, when known. The primary benchmarks for six tide gauges (Caminada Pass, East Bay, Freshwater Canal Locks, Grand Isle, Lafitte, and Martello Castle) are all set into some type of concrete structure rather than attached to steel rods that are driven to depth. Similar to the argument outlined above for the seawall at Grand Isle, we reason that most, if not all, of the concrete structures hosting benchmarks in coastal Louisiana likely have some type of foundation that extends below the ground surface. If this is the case, the primary benchmarks for the six tide gauges listed above should not be considered surface marks, although their foundations may be considerably shallower than the deepest benchmarks.

The benchmark datasheets available on the NOAA website provide only basic descriptions of the concrete structures on which these primary benchmarks are mounted: concrete retaining wall (Caminada Pass), concrete platform (East Bay), cement structure (Freshwater Canal Locks), concrete seawall (Grand Isle), concrete foundation for a small pumping station (Lafitte), and rough poured concrete (Martello Castle).

Note that the primary benchmark for the Weeks Bay tide gauge is attached to a steel rod driven to an unspecified depth, as is the primary benchmark at Cypremort Point. We assume that these rods are similar in length to those used for other NOAA benchmarks (~10-35 m) and thus these benchmarks should not be considered surface marks.

**Just in case, I checked my interpretation about the way NOAA works with the COOPS Technical Director (Dr. Peter Stone) and Chief Scientist (Dr. Greg Dusek). They replied: "We control the water level observation primarily off of one primary bench mark (PBM) and then ensure the stability of that mark by using the remaining 9 or so marks. On occasions when we see substantial and/or continual differential movement between the PBM and the other marks, we adjust the PBM to a different mark determined to be**
**stable relative to the remaining marks."**

We appreciate the input from Dr. Peter Stone and Dr. Greg Dusek. We have added this clarification to the manuscript: Tide gauges are typically leveled using a benchmark designated as the primary benchmark; secondary benchmarks are used to assess the stability of the primary benchmark (NOAA, 2013). See lines 95-97 in the manuscript.

**So that confirms what is on the NOAA web site, and confirms that my comments about the six mentioned above, and Grand Isle in particular, are correct. They do not fit into the main argument of the paper, so there should be some extra wording to handle that. As for the other 25 stations in Table 1 for which the primary mark is a deep one, then I agree with their comments, but only in principle, and only at a time way into the future when these stations will have acquired records long enough for trend estimation.**

For our original analysis, we chose to use the benchmark with the deepest known foundation in order to maximize the size of our dataset: 35 tide gauges have at least one benchmark with known foundation depth, but primary benchmark depths are known for only 23 tide gauges. Based on the reviewer's thoughtful comments, we have added an analysis of primary benchmark foundation depths. For benchmarks with known foundations depths (i.e. those mounted on steel rods driven to refusal), we find that primary and deepest known benchmarks are anchored an average of $21.4 \pm 3.9$ m and $21.5 \pm 7.4$ m below the surface, respectively. Note that for 8 of 23 tide gauges (35%), the primary benchmark is also the benchmark with the deepest known foundation. The mean foundation depth for all benchmarks is $21.0 \pm 5.4$ m. Thus, we see that primary benchmark foundation depths are indistinguishable from the dataset as a whole. We have added this new analysis in lines 190-196 in the manuscript.

As discussed above, benchmarks anchored on concrete structures are unlikely to have a foundation depth of zero. Instead, we suggest that the concrete structures that host benchmarks are likely anchored at some depth below the surface and thus the associated tide gauges continue to support the main argument of our paper: that tide gauges with benchmarks anchored at depth do not record all shallow subsidence. We do recognize, however, that these structures with unknown exact foundation depths, may be anchored at shallower depth than steel rods. See expanded discussion of this issue in lines 183-188 and 235-248 of the manuscript.

**Stone and Dusek remarked: "The large number of tide gauges used in the analysis is very perplexing. The NOAA gauges [mentioned in Table 1] (which were installed by CO-OPS) were installed for wide ranges of time. Two of the gauges (Shell Beach and Grand Isle) were installed for decades and we have calculated relative sea level change rates. The others have only been installed for a few months or years and do not have enough data to calculate statistically significant RSLR [relative sea level rise]."**

It is true that many of the tide gauges used in our analysis of benchmark depths were active for only a couple of years and do not have enough data to calculate a meaningful rate of RSLR. However, the purpose of this study is to investigate the foundation depths of published benchmarks, not to re-calculate rates of RSLR. Even though a significant proportion of the tide gauges may never become suitable for RSLR studies, they allow us to greatly expand the dataset on benchmark foundation characteristics. All of the foundation depths discussed in the paper come from currently-published benchmarks, even if the associated tide gauges are no longer active. See lines 222-224.

**Now, Grand Isle I have already mentioned. In fact, Shell Beach has a deep primary mark, so I accept that the argument of the authors applies for that. But as Shell Beach has data (in the PSMSL) only for 2008-2017, it is hardly yet a long record. So I think some care should be taken in**

**the text between explaining what could happen IN PRINCIPLE regarding tide gauges with deep primary marks, and what is the real situation at the moment in the delta.**

This is a good point. We have added clarification that many of the tide gauges listed in Table 1 are not useful for RSLR analyses due to their short records. However, some of the tide gauges that currently have short records could become important in the future as their records become longer (e.g. Shell Beach). Additionally, our analysis shows that benchmarks with deep subsurface foundations are the norm in coastal Louisiana and thus any rates of RSLR calculated using tide-gauge data likely do not include shallow subsidence. See lines 224-231.

**This takes me to two mentions of the PSMSL in the paper. At line 50 the authors state that there are 5 PSMSL stations in Lousiana but do not give their names. They are Eugene Is (data 1939-1974), Grand Isle (1947-2017), South Pass (1980-1999), Shell Beach (2008-2017) and New Canal Station (2006-2017). As mentioned above, Grand Isle is the only important one for sea level trends. The PSMSL defines RLR datum at Grand Isle (and other NOAA sites) using the Station Datum information in each case that NOAA provides. Therefore, I think there should be a mention somewhere in the paper to the effect that the sea level rate at Grand Isle provided by the PSMSL record is not likely under-estimated as the text presently implies.**

We have added the names of the five tide gauges and the years for which they have produced relative sea-level data (see lines 74-75). Our results continue to suggest that the Grand Isle tide gauge is likely underestimating the rate of RSLR. See detailed discussion above.

**The other mention of the PSMSL is in the paragraph at lines 250-261. It again mentions only 5 PSMSL stations in the area. Why? The PSMSL cannot be expected to databank the density of stations that the authors need, so to somehow conflate the PSMSL with that requirement seems strange to me. In fact, what the PSMSL would be happy with in an area this size is a single tide gauge station with GPS and good BM control. Anyway, the authors show potentially they have many more than 5 so what is their problem? Also the paragraph says that of the 5 'only a few' have RSET nearby. If one takes 'a few' as meaning 3 or similar then one could read this sentence as saying that most PSMSL stations have RSET, which I think is opposite to what the authors want to say! So this paragraph needs rewording and I can't see why the PSMSL is being dragged into it at all.**

We agree that conflating PSMSL with our proposed network is not appropriate. We have deleted the mention of PSMSL in this paragraph and reworded the text accordingly.

**Conclusions - so I see the problems that the authors raise about deep BMs, in principle. However, I do not buy the suggestion that, instead of tide gauges, a better job could be done using RSET-MH data which seems to me to be a very rough and ready method, combined with GPS for deep submergence, combined with altimetry. RSET, GPS and altimetry data all have their own nuances and problems, and in particular altimetry until fairly recently has had problems getting very close to the coast. Tide gauges could do the job you want if you have at least one surface mark at each site, and if there is ongoing monitoring of the relative heights between surface and deep BMs. That**

**would solve your problem; a conversation with NOAA is required about constructing a history of the evolution of relative heights between benchmarks at each site.**

We have added to the section in the manuscript acknowledging the limitations of various instruments and methods of measuring RSLR and describe data analysis techniques that could be used to overcome some of these shortcomings (see lines 330-346). As discussed above, however, we believe that most, if not all, benchmarks in coastal Louisiana are anchored at depth, including those anchored on concrete structures. There is no evidence for any true surface benchmarks in the area.

**Stone and Dusek commented to me also that "The authors' did not address the mounting of the water level sensors on different structures they were comparing and how those structures can be affected by settling. Some of the stations used in their comparisons are probably installed on piers where the pilings may only be sunk a few feet. Others, like the water level stations at Shell Beach and Calcasieu Pass, LA are mounted on massive steel structures driven into consolidated sediments (which we refer to as SPIPs). The type of installation can be relevant to consider when attempting to accurately assess sensor movement relative to bench marks on land, and presumably in the cases of the SPIPs, our leveling data could indicate if deep rod marks and the shallower rod or concrete marks show variable long-term trends relative to the water level observations."**

As we understand, this is exactly why tide gauges are leveled using benchmarks in the first place: to correct for any drift in the instrument that could be caused by a variety of processes, including the settling of the support structure. It seems that this issue is accounted for in well-executed monitoring programs, and thus is not something that our study needs to consider.

**So, while accepting the general main point of the authors, I think the main thing is to have access to histories of all the relevant surveying information at a site. A last comment about the Conclusions is an obvious one, that the correct scientific approach is to make use of data from all techniques and see eventually how they compare, not just suggest rejecting tide gauges (which NOAA pay for, given that they are anyway needed for monitoring transient events such as storm surges) by adopting an 'alternative approach'.**

This is an excellent point. We now note that best scientific practices will make use of all available data and compare the results of various measurement techniques. Furthermore, tide gauges remain critical for measuring many processes, including tides (the original and still-primary purpose of tide gauges) and event-scale phenomena such as storm surge, and are invaluable in this regard. See lines 311-315.

**The Conclusions also makes some comments about deltas elsewhere around the world and lists some in Table 3. How many have deep BMs like in Louisiana? I suspect most do not, but at best have surface marks in which these arguments will not apply. It would be interesting to know.**

We believe that benchmarks in other low-elevation coastal zones are likely constructed in a broadly similar fashion to those in coastal Louisiana: either attached to rods driven to refusal or mounted on existing structures with non-negligible foundation depths. For example, from conversations with Dutch colleagues, we understand that tide-gauge benchmarks in The Netherlands are ~5-25 m deep and generally anchored in the Pleistocene basement except in areas very near the coast where the Pleistocene sediment thickness is greatest (See Table 4). In other words, conditions in The Netherlands are roughly comparable to those in the Chenier Plain of coastal Louisiana (and likely other "thin" LECZs): they do not capture the shallow subsidence component, but since benchmarks are generally anchored in a relatively stable substrate they are easier to interpret than many of the tide gauges in the Mississippi Delta (and likely other "thick" LECZs) where benchmarks are essentially "floating" in the Holocene succession. Although we are fortunate to have acquired precise benchmark data from The Netherlands, we have found that information on benchmarks in other LECZs is very difficult to come by. A global analysis of benchmark construction would be a valuable but massive undertaking and is beyond the scope of the present manuscript.

**So for the reasons above I think some rewriting of the text is required.**

**Detailed comments:**

**line 17 and elsewhere - GPS is better denoted at GNSS (Global Navigation Satellite System) these days.**

Thank you for this recommendation. We have changed GPS to GNSS throughout the manuscript.

**line 35 - a reference to long tide gauge records in N Europe and N America could be Woodworth et al. (Surveys in Geophysics, 2011). The longest US record was claimed for many years to be Key West (Maul and Martin, GRL, 1993) but I guess now one should also mention Boston (Talke, JGR, 2018).**

We have added mention of the Key West, Boston, and San Francisco tide gauges and references to Maul and Martin (1993), Woodworth et al. (2011), and Talke et al. (2018). See lines 56-58.

**line 49 - the PSMSL should be referenced by its web site and journal (http://www.psmsl.org and Holgate et al., J Coastal Res, 2013)**

We have added these references. See lines 72-73.

**line 50 - give the names of the five (see above)**

We have added the names of the five tide gauges and the years for which they have relative sea-level data. See lines 74-75.

**line 89 - these references should also include the IOC Manuals, see http://www.psmsl.org/train_and_info/training/manuals/**

Good suggestion, we have added a link to the IOC training manuals. See lines 124-125.

**line 125 - a reference is needed for where you got the Pleistocene surface information from.**

We have added a reference to Heinrich et al. (2015). See line 165.

**line 169 - 'because all tide gauge benchmarks'. This is not true, see above.**

We have clarified that all tide gauge benchmarks with KNOWN foundation information are anchored at depth. See lines 226-227.

**line 172 - I would be grateful if you did not use the word 'eustatic' which means different things to different people (there is a recommendation about this in one of the IPCC reports). I suggest this is reworded: ... deep subsidence plus the component of RSLR associated with changes in real ocean level. (or something like that). And drop 'as well .... effects'.**

Thank you for this suggestion. We now say, "…deep subsidence plus the component of RSLR associated with changes in real (geocentric) ocean level…". See lines 229-230.

**line 186 - I would reword: ... includes subsidence of that part of ... sediments deeper than the BM depth.**

Good suggestion. We now say, "…deep subsidence also includes subsidence of the part of the Holocene sediment column that underlies the benchmark foundation." See lines 265-267.

**line 207 - reword to avoid eustatic: .. adding the historic rate of real (geocentric) sea-level rise ..**

We have made this change throughout the manuscript.

**There is a reference to Ericson et al. (2006) in the context of not using tide-gauge data. But the sea level rise value in that paper was just the global average taken from the IPCC (1.5 mm/yr) which hardly seems to me to be superior to using local tide gauges where available. I realise why Ericson et al. had to do that in their paper but it is not to be recommended in your case.**

Thank you for pointing out this limitation of the Ericson et al. (2006) method of assessing delta vulnerability. We have added a note in the manuscript that this approach is hindered by relying on measurements of global rather than local sea-level rise (see lines 301-302). We have chosen to keep the reference to Ericson et al. (2006), however, because we feel it is an important example to include in our discussion of previous studies of delta vulnerability that did not use tide-gauge data.

**line 223 - a reference is needed for the InSAR mention, preferably for its use in deltas.**

We have added references to Dixon et al. (2006), Jones et al. (2016), and Da Lio et al. (2018). See lines 341-342.

**230-233 - SWOT is only one of several efforts to improve coastal altimetry. A general reference, in which there is mention of SWOT, would be: Vignudelli, S., Kostianoy, A., Cipollini, P and Benveniste, J. (eds). 2011. Coastal altimetry. Berlin: Springer Publishing. 578pp.**

Good to know. We have added the note that SWOT is one of several efforts currently in the works and we now cite Vignudelli et al. (2011). See lines 332-336.

**237 - 'commonly too noisy'. From what I have read of the method I'm not surprised!**

A single RSET-MH produces noisy shallow subsidence data because several spatially and temporally variable processes affect wetland surface elevation (e.g. tidal stage, wind direction and strength, belowground biomass). Despite this limitation, robust measurements of shallow subsidence can be produced by an RSET-MH dataset that includes measurements from numerous RSET-MHs. See discussion of this issue in lines 347-374.

**References - need doi's adding**

DOIs are not required for submission to Ocean Science, so we have left them off for now. If required by the journal, we would be happy to add them later.

**453 - there should be an accent over the 'i' in Miguez**

Corrected, thank you.

**Figure 1 - What are the short and long vertical lines beneath the tide gauge in each panel supposed to be showing. I'd remove them. The point is that the datum of a tide gauge is determined by levelling to the BM nearby, so I would have the horizontal red line for the tide gauge at the same level as the BM and a dotted line between them. I think this figure may have been adapted from Figure 1 of Webb et al. (2013) which in their case has a short vertical band which I think is supposed to be indicating a float gauge (of which there are fewer around), and a longer vertical band which I think has the same function as the vertical red line for the BM in the present case. Anyhow, please lose the vertical lines under the tide gauges in this case.**

In Figure 1, each tide gauge is now represented by a narrow red rectangle and is connected to the benchmark with a dashed line.

**Figure 2 and 3 - could the lat/lon ticks and annotation face outside the map to be clearer?**

These changes have been made.

**Figure 3 - could the colour scale on the right be labelled Pleistocene Depth and the insert headed FD above PS or similar? As mentioned above, a reference is needed in the caption for the Pleistocene depth information. The black lines for the shoreline are hard to see. There is no point**

**mentioning ENG1 and 2 if they don't appear on the plot. But perhaps say (ENG1 andd ENG2, see Table 2).**

In Figure 3, labels have been added to the color bar and to the inset box. In the caption, we have clarified that the Pleistocene depth information is from Heinrich et al. (2015). As also suggested, we now say "ENG1 and ENG2, see Table 2". We found that a thicker black line indicating the shoreline becomes distracting, so we have left the shoreline as-is.

**Figure 4 - nice plot.**

Good to hear, thank you.

**Figure 6 - put mm/yr after mean, standard deviation**

We have made this change.

**Tables 1 and 2 - head the column 'Maximum benchmark foundation depth (m)'**

We have made this change to Table 1. We did not make this change to Table 2 because the table refers to GNSS stations, which are not associated with benchmarks. The foundation depths listed in Table 2 indicate the depth of the rod or structure on which the GNSS station is mounted.

**Figure S1 - I don't see the point of this figure. It has no more information than Figure 6. Doesn't do any harm I guess.**

We have removed Figure S1 to avoid redundancy.

**I hope these comments are useful. I have no objection to my identity being revealed. I am very grateful to Peter Stone and Greg Dusek for information which helped me complete this review.**

Thanks again to all three for these very thorough and thoughtful comments.

Response to the manuscript review by an anonymous referee
Reviewer comments are in bold text; author responses are in normal text
* * *
**General comments:**

**This study seems closely related to the work published in GSA Today (Nienhuis et al. 2017) in which the authors were involved. But it is not clearly stated how both relate together. Nienhuis et al. is quoted towards the end of the manuscript, just before the conclusions. The findings on the underestimation due to shallow subsidence are already present in Nienhuis et al. Hence, should this manuscript be considered as supplemental information to the GSA Today one? I think it is important that the authors clarify how both studies relate together from the beginning (introduction). In addition, the introduction and the conclusion ("we present", "we propose") suggest the approach is novel. However, later on we find expressions and references which suggest it is not. Overall the authors need to make an effort to unambiguously set their study in the scientific context.**

First, we would like to thank the anonymous referee for the thoughtful feedback regarding our manuscript. We have taken the referee's suggestions into account and feel that it has enabled us to make significant improvements.

The reviewer is correct in noting that there is a brief mention in the Nienhuis et al. (2017) paper that benchmarks in coastal Louisiana are typically anchored at depth and thus the associated tide gauges do not capture shallow subsidence. However, Nienhuis et al. do not go into any detail about how this information was acquired or methods to remedy this issue. Instead, the paper is relatively narrowly focused on presenting a subsidence map for coastal Louisiana and it is not concerned with the methodology of measuring present-day RSLR in LECZs in a more general sense. In the present manuscript, we present and analyze benchmark depth data, discuss limitations of a variety of techniques for measuring RSLR, and suggest an alternative method of measuring RSLR in LECZs. While the scope of the Nienhuis et al. paper is strictly limited to coastal Louisiana, here we use coastal Louisiana as a case study for an issue that is likely global in scope. Thus, we hope to reach a much wider audience than the target audience for the Nienhuis et al. paper. Therefore, while the reviewer is correct that there are distinct elements that connect the two studies, these two manuscripts are otherwise separate and stand alone. We have clarified this connection in lines 117-120 of the manuscript.

As for the novelty of our manuscript, the practice of using RSET-MHs to measure shallow subsidence is not new and we cite two studies using state-of-the-art RSET-MH methods: Webb et al. (2013) and Cahoon (2015). What is novel is the method of combining RSET-MH data with data from GNSS stations and satellite altimetry in order to produce robust measurements of RSLR. This method was first introduced by Jankowski et al. (2017), but for a different purpose (to evaluate the ability of coastal wetlands to keep pace with RSLR). Here we explore this new approach in much more detail and with the explicit objective to reach the large, multidisciplinary community concerned with obtaining better measurements of present-day rates of RSLR. We now clarify these points in lines 117-120 and 138-140 of the manuscript.

**In my opinion, the manuscript suffers from a perspective bias "against" tide gauges. That is, the authors show that both the tide gauges and GPS antennas are similarly anchored deep below the surface (at almost equivalent depths). Thus, none of them can actually capture the shallow**

**subsidence. The combination of satellite altimetry and GPS data or the use of tide gauges suffer from the same drawback. Consequently, the statement that the novel approach eliminates the need for tide gauge data (repeated several times in the manuscript) is not objectively supported, because the same criticism applies to GPS antennas, and hence to the combination of satellite altimetry and GPS data. From my understanding, tide gauges + RSET-MH can work as well as satellite altimetry + GPS + RSET-MH. The authors need to think about it, and provide arguments to support their claim in a more convincing way, or reconsider the presentation of their findings (which are anyway interesting, in my opinion).**

This is an excellent point. We have adjusted our wording throughout the manuscript to clarify that tide gauges are critical for many applications and that we are merely discussing a specific (yet important) context where tide-gauge data may not be the best option. In the abstract and conclusions (see lines 43 and 392), we now say that our proposed method of measuring RSLR in LECZs eliminates the need for tide-gauge data "in this context". Tide gauges remain critical for measuring many processes, especially tides (the original and still-primary purpose of tide gauges) and event-scale phenomena such as storm surge, and they are invaluable in this regard. We also note that best scientific practices will make use of all available data and compare the results of various measurement techniques. See lines 311-315.

Indeed, many of the issues affecting tide gauges also affect GNSS stations. Both types of instruments are generally anchored at depth and thus do not capture shallow subsidence. In principle, both GNSS stations and tide gauges could be used to measure deep subsidence and these data could then be combined with measurements of shallow subsidence (plus geocentric sea-level rise, in the case of GNSS data) to calculate RSLR. However, the tide gauges must have sufficiently long time series (at least 30 years) and known foundation depths to be useful in this context. In coastal Louisiana, the number of tide gauges that meet these criteria ($n = 5$) are fewer in number than GNSS stations with known foundation depths ($n = 10$). Additionally, GNSS data are less susceptible to short-term environmental conditions (i.e. wind speed and direction, tides, atmospheric pressure changes) than are tide gauge data. Thus, GNSS is the preferred method for measuring deep subsidence. This additional information is now included in lines 316-321 and 327-329 of the manuscript.

**The manuscript (introduction) suggests an assessment of their findings in LECZs worldwide, but the authors do not provide evidence that the findings apply beyond their case study zone, except for some general considerations (sediment thick in different coastal areas of world from the literature). The authors should be aware that different countries (agencies) have different practices in building infrastructures (tide gauges or GPS antenna monumentations). The US case study is likely not representative of the wide range of practices elsewhere. They should consider reducing the scope of the claims, and develop a cautious discussion in extending the findings in LECZ worlwide. The title may be revisited too.**

We have acquired information on benchmarks in The Netherlands and now include them in the manuscript for comparison. From conversations with Dutch colleagues, we understand that tide-gauge benchmarks in The Netherlands are ~5-25 m deep and generally anchored in the Pleistocene basement except in areas where the Holocene sediment thickness is greatest (see newly-added Table 4). In other words, conditions in The Netherlands are roughly comparable to those in the Chenier Plain of coastal Louisiana (and likely other "thin" LECZs): they do not capture the shallow subsidence component, but because benchmarks are generally anchored in a relatively stable substrate they are easier to interpret than many of the tide gauges in the Mississippi Delta (and likely other "thick" LECZs) where benchmarks are essentially "floating" in the Holocene succession. This additional information is included in lines 279-

287. We expect that benchmarks in other LECZs are likely constructed in a broadly similar fashion to those in coastal Louisiana and The Netherlands: either attached to rods driven to refusal or mounted on existing structures with non-negligible foundation depths. Although we are fortunate to have acquired relatively precise benchmark data from The Netherlands, we have found that information on benchmarks in other LECZs is very difficult to come by. A global analysis of benchmark construction would be a valuable but massive undertaking and is beyond the scope of the present manuscript.

**In line with the above comment, I would suggest a search in the literature about GPS station monumnetations to support the worldwide extension. I vaguely remember a talk a decade ago or so about GPS antenna monumentations within an IGS meeting or an IAG scientific assembly. The concern of the study was the ability of the different types of GPS antenna monumentations to estimate actual ground / crustal motions. I think it might be worth searching for the details of this study or later studies on this subject.**

Concerted efforts are currently underway to address the complexities regarding GNSS monumentation. At a newly-constructed subsidence superstation located in the lower Mississippi Delta, for example, three GNSS instruments were anchored at different depths in order to build a depth-integrated profile of subsidence (Allison et al., 2016). Novel approaches like these are expected to greatly improve our understanding of subsidence in LECZs in the future. This information is now mentioned in lines 322-326.

In addition, we now refer to the information available on hundreds of individual GNSS stations through the International GNSS Service (http://www.igs.org/network) and the National Geodetic Survey (https://www.ngs.noaa.gov/CORS/). See lines 127-129 in the manuscript. Site photos indicate that most GNSS stations are indeed mounted on existing buildings. Although the foundation depth of these buildings likely varies and tracking down foundation information for each building would require an enormous effort, it is likely that most (if not all) are anchored at some depth beneath the surface. Put differently, it is unlikely that these buildings are simply floating on the ground surface.

**In addition to the above comment, the choice of the deepest benchmark in section 4 needs to be supported, especially regarding the leveling analysis and practice to maintain the tide gauge datum, which can differ from one country (agency) to another (agency). Furthermore, I think this methodological choice should not be presented / discussed in the "Results" section but in the methods section.**

Excellent point. For our original analysis, we chose to use the benchmark with the deepest known foundation in order to maximize the size of our dataset: 35 tide gauges have at least one benchmark with known foundation depth, but primary benchmark depths are known for only 23 tide gauges. For comparison, we have added an analysis of primary benchmark foundation depths. For benchmarks with known foundations depths (i.e. those mounted on steel rods driven to refusal), we find that primary and deepest known benchmarks are anchored an average of $21.4 \pm 3.9$ m and $21.5 \pm 7.4$ m below the surface, respectively. Note that for 8 of 23 tide gauges (35%), the primary benchmark is also the benchmark with the deepest known foundation. The mean foundation depth for all benchmarks is $21.0 \pm 5.4$ m. Thus, we see that primary benchmark foundation depths are indistinguishable from the dataset as a whole. We have improved the explanation of our methods (and agree that it fits better in the Methods section than in the Results) and added a description of this new analysis in lines 167-170 and 190-196 in the manuscript.

In addition, we have acquired information on benchmarks in the Netherlands (see above for an in-depth discussion). Dutch benchmarks are constructed in a similar fashion as those in coastal Louisiana (i.e.

mounted on steel rods, sheet piling, or concrete) and are also anchored at depth. Foundation depths range from 5 to 25 m. This information is now included in the manuscript in lines 279-287 and in Table 4.

**The manuscript is overall well written with good illustrations (Figures). In my opinion, it needs to consider the above comments. My suggestion is therefore a major revision.**

**Specific comments & Technical corrections:**

**p.2, L17-18: The expression is confusing. That is, if the station is >14 m, it includes the surface, and thus can capture any land motion. Consider rephrasing, why not using the same form as with the tide gauges? (Simply remove ">").**

We compiled GNSS station foundation information from Dokka et al. (2006) and Karegar et al. (2015). In these papers, minimum (rather than exact) foundation depths are given for two of the GNSS stations. They are reported as >20 m (site BVHS) and >15 m (site HOUM) and we adopt this notation in our manuscript (see Table 2). We have now highlighted these sites more clearly in Figure 4. We use these minimum foundation depths for the BVHS and HOUM stations when calculating the mean foundation depth for all GNSS stations and then indicate that this mean value is in fact a minimum value. In line 208 (see also lines 39 and 377), we report that GNSS stations are anchored an average of >14.3 m below the land surface (i.e. the average foundation depth is no shallower than 14.3 m) and thus do not include processes occurring in the shallow subsurface.

**p.2. L22: the need for tide gauge data is often multi-application. The authors should be careful with this claim, and state the context of it (eliminates the need for this specific application and LECZ situation). In addition, see major comment above, that is, the same concerns apply to the GPS monumnetation, hence both tide gauges and GPS show the same drawback.**

Please see above for an in-depth discussion of this issue. We have adjusted our wording throughout the manuscript to clarify that we are focused on a specific context where tide-gauge data may not be the best option. Tide gauges remain critical for measuring many processes and are invaluable in this regard. See lines 311-315.

**p.2, L34: a reference to support this claim is missing. It could be Holgate et al. (2013) which describes a data bank or similar; it could be an (the) article(s) that rescued the historical data of the stations listed in brackets.**

We now refer to five of the longest tide gauge records and cite three papers that presented the historical data: Key West, USA (Maul and Martin, 1993); Brest, France; Świnoujście, Poland; New York, USA; and San Francisco, USA (Woodworth et al., 2011); Boston, USA (Talke et al., 2018). See lines 55-58.

**p.2. L37: Watson et al. is a good paper but it is not relevant in the context of this sentence. (Its global sea-level rise estimate is based on satellite altimetry data). Maybe the reference can be used somewhere else.**

We have removed the reference to Watson et al. (2015) from this sentence.

**p.3 L50. Consider adding the reference for the PSMSL (Holgate et al. 2013 in J. Coastal Res).**

Good suggestion, we have added a reference to Holgate et al. (2013) as well as the PSMSL web address (http://www.psmsl.org). See lines 72-73.

**p.3. L53. What signals encompass "natural variability" here?**

In lines 78-81, we have clarified that the natural variability includes phenomena such as storms, El Niño-Southern Oscillation cycles, changes in the orbital declination of the moon, shifts in ocean currents, and atmospheric pressure variability (Pugh, 1987; Douglas, 1991; Shennan and Woodworth, 1992).

**p.4, L93. A reference is needed to support this claim. I vaguely remember a talk several years (decade?) ago at an IGS or IAG meeting about GPS antenna monumentations (structure, depth. . .) with some statistics. The concern of the study was the ability of GPS antennas to estimate actual ground / crustal motions. I think it can be worth searching the literature, especially since L97 states the issue of the nature of GPS station foundations as an objective of the study.**

Please see above for an in-depth discussion of this topic. Efforts are currently underway to address the complexities regarding GNSS monumentation. In addition, we now refer to the information available on hundreds of individual GNSS stations through the International GNSS Service and the National Geodetic Survey.

**p.4. L100. Confusing (see general comments above). The expression suggests the approach is novel, especially because in the previous sentence it is stated what is not the purpose of the study. However, there are two references at the end of the sentence. Is this study a refinement? Consider rephrasing and clarifying.**

Please see above for an in-depth discussion of this issue. In this manuscript, we present a novel method to measure RSLR in LECZs. We now clarify that the reader should see Webb et al. (2013) and Cahoon (2015) for descriptions of the RSET-MH method (see lines 138-140), which can be used to measure one component of RLSR (shallow subsidence).

**p.5, L144-145. This choice needs to be supported, especially regarding the leveling analysis and practice to maintain the tide gauge datum, which can differ from one country (agency) to another (agency).**

Please see above for an in-depth discussion of this topic. In our original analysis, we chose to use the benchmark with the deepest known foundation in order to maximize the size of our dataset. For comparison, we have added an analysis of primary benchmark foundation depths. We find that primary benchmark depths are indistinguishable from the dataset as a whole. We have improved the explanation of our methods and added a description of this new analysis in lines 167-170 and 190-196 in the manuscript.

For a better global context, we now include information on benchmarks in the Netherlands, which are constructed in a similar fashion as those in coastal Louisiana and are also anchored at depth. Discussion of this information is now included in the manuscript in lines 279-282.

**p.6, L172 (L205 too). What is behind the term 'eustatic'?**

The term "eustatic" has been removed from the manuscript and replaced with clearer terminology. We now refer to this phenomenon as "real (geocentric) sea-level rise".

**p.14. I cannot see whether there are squares and circles co-located. Consider using a different colour too, it may help.**

The color scheme in Figure 2 has been changed to improve readability. Tide gauges and GNSS stations are now shown as dark blue circles and light orange squares, respectively.

[revised manuscript text omitted]

Allison, M., B. Yuill, T. Törnqvist, F. Amelung, T.H. Dixon, G. Erkens, R. Stuurman, C. Jones, G. Milne, M. Steckler, J. Syvitski, and P. Teatini. 2016. Global risks and research priorities for coastal subsidence. Eos, 97 (19), 22-27.

Amorosi, A., L. Bruno, D.M. Cleveland, A. Morelli, and W. Hong. 2017. Paleosols and associated channel-belt sand bodies from a continuously subsiding late Quaternary system (Po Basin, Italy): New insights into continental sequence stratigraphy. Geological Society of America Bulletin, 129, 449-463.

Cahoon, D.R. 2015. Estimating relative sea-level rise and submergence potential at a coastal wetland. Estuaries and Coasts, 38, 1077-1084.

Cahoon, D.R., D.J. Reed, and J.W. Day, Jr. 1995. Estimating shallow subsidence in microtidal salt marshes of the southeastern United States: Kaye and Barghoorn revisited. Marine Geology, 128, 1-9.

Church, J.A. and N.J. White. 2011. Sea-level rise from the late 19th to the early 21st century. Surveys in Geophysics, 32, 585-602.

Church, J.A., P.U. Clark, A. Cazenave, J.M. Gregory, S. Jevrejeva, A. Levermann, M.A. Merrifield, G.A. Milne, R.S. Nerem, P.D. Nunn, A.J. Payne, W.T. Pfeffer, D. Stammer, and A.S. Unnikrishnan. 2013. Sea Level Change. In: T.F. Stocker, D. Qin, G.-K. Plattner, M. Tignor, S.K. Allen, J. Boschung, A. Nauels, Y. Xia, V. Bex, and P.M. Midgley, eds., Climate Change 2013: The Physical Science Basis. Contribution of Working Group I to the Fifth Assessment Report of the Intergovernmental Panel on Climate Change. Cambridge University Press, New York, NY, USA, 1137-1216.

Cipollini, P., F.M. Calafat, S. Jevrejeva, A. Melet, and P. Prandi. 2017. Monitoring sea level in the coastal zone with satellite altimetry and tide gauges. Surveys in Geophysics, 38, 33-57.

Clift, P.D., L. Giosan, A. Carter, E. Garzanti, V. Galy, A.R. Tabrez, M. Pringle, I.H. Campbell, C. France-Lanord, J. Blusztajn, C. Allen, A. Alizai, A. Lückge, M. Danish, and M.M.

Rabbani. 2010. Monsoon control over erosion patterns in the Western Himalaya: possible
feed-back into the tectonic evolution. Geological Society Special Publication, 342, 185-
218.

Da Lio, C., P. Teatini, T. Strozzi, and L. Tosi. 2018. Understanding land subsidence in salt
marshes of the Venice Lagoon from SAR interferometry and ground-based
investigations. Remote Sensing of Environment, 205, 56-70.

Day, J., C. Ibáñez, F. Scarton, D. Pont, P. Hensel, J. Day, and R. Lane. 2011. Sustainability of
Mediterranean deltaic and lagoon wetlands with sea-level rise: The importance of river
input. Estuaries and Coasts, 34, 483-493.

Dixon, T.H., F. Amelung, A. Ferretti, F. Novali, F. Rocca, R. Dokka, G. Sella, S.-W. Kim, S.
Wdowinski, and D. Whitman. 2006. Subsidence and flooding in New Orleans. Nature,
441, 587-588.

[revised manuscript text omitted]